# Fine-Tuning of Transformer models with Frames

**Harshavardhan Adepu** [1]  **Li Zhang** [2]  **Sanjiv Kumar** [2]  **Vikas Singh** [1]

## Abstract

Parameter-Efficient Fine-Tuning (PEFT) strategies such as Low-Rank Adaptation (LoRA) are effective solutions for fine-tuning large-scale pretrained models; however, their memory requirements scale with the size of the model, $\mathcal{O}(dr)$, where $d$ is the model's hidden dimension and $r$ is the rank. Our proposal, FrameFT, models the parameter update $\Delta W$ with a sparse coefficient matrix in a Fusion Frame basis. Fusion Frames can be generated *algorithmically* and shared across model layers, enabling very efficient updates. Only the sparse coefficients of the basis expansion are stored/optimized, reducing the memory footprint. The sparse structure of the coefficient matrix in FrameFT *and* the sparsity in the Fusion Frames give large compute benefits, and our analysis provides formal convergence results. We evaluate the idea across a suite of supervised fine-tuning benchmarks, focusing on language tasks, but also report application to vision models. Our experiments show that FrameFT achieves performance on par with/exceeding state-of-the-art PEFT techniques, but needs far fewer trainable parameters.

## 1. Introduction

Foundation models show impressive capabilities across a range of domains, including language (Touvron et al., 2023; Team et al., 2024a; OpenAI, 2025), vision (Dehghani et al., 2023), and other modalities (Łajszczak et al., 2024; Fang et al., 2025). While increasing model size gives improved performance, many downstream applications benefit from an additional fine-tuning step to specialize the model for specific tasks. However, fine-tuning all parameters of a large model is expensive. This becomes worse when fine-tuning is required for multiple tasks, each needing to store its own dedicated copy of the model. To address these difficulties, Parameter-Efficient Fine-Tuning (PEFT) methods provide an alternative (Hu et al., 2022; Zhang et al., 2025; Dettmers et al., 2023). These techniques seek to minimize the number of trainable parameters during fine-tuning, significantly reducing resource needs. Recent work has shown that PEFT methods can match the performance of full-model fine-tuning while updating only a small subset of the parameters (Bensaïd et al., 2025; Zhang et al., 2026).

**LoRA and Sparse Fine-tuning.** The most widely used PEFT technique is Low-Rank Adaptation (LoRA) (Hu et al., 2022), which keeps the pre-trained weights frozen and injects task-specific low-rank matrices into each layer. These matrices provide the flexibility to adapt the model while maintaining low memory/compute costs. LoRA's success has inspired a range of variants that offer improvements in convergence, efficiency, and storage overhead (Dettmers et al., 2023; Hayou et al., 2024; Zhu et al., 2024; Yaras et al., 2024; Xia et al., 2024). One line of work uses the Singular Value Decomposition (SVD) of pre-trained weights to construct fixed bases for adaptation (Lingam et al., 2024; Bałazy et al., 2025). However, this needs dense singular vectors, which increases the memory footprint and affects inference throughput. This overhead is problematic in multi-tenant serving scenarios where there is a single pretrained model and its many fine-tuned versions.

In contrast, sparse fine-tuning (SFT) methods use a different strategy: they selectively update a sparse subset of the model's original parameters. The selection rule varies: some approaches rely on magnitude-based pruning (Lu et al., 2024), while others use sensitivity measures such as Fisher information (Guo et al., 2021), Fourier-domain analyses (Gao et al., 2024), or parameter change measures (Ansell et al., 2022). SFT methods can face scalability issues when applied to large models, especially in identifying sparse patterns and tuning hyperparameters (Guo et al., 2021). Some proposals also rely on access to training dynamics (Ansell et al., 2022) that may not be easily available.

**Motivation.** LoRA's dense singular vectors and asymmetric $A/B$ structure were inherited from matrix approximation methods and shown to work well for fine-tuning. SFT methods similarly inherited ideas from model pruning,

---

[1]University of Wisconsin-Madison [2]Google DeepMind. Correspondence to: Harshavardhan Adepu <adepu@wisc.edu>.

*Proceedings of the 43$^{rd}$ International Conference on Machine Learning*, Seoul, South Korea. PMLR 306, 2026. Copyright 2026 by the author(s).

i.e., identify task-relevant parameters through gradient or magnitude-based proxies. We instead take a use case driven approach and ask: how should we parameterize weight updates if we want expressivity, parameter efficiency, stable optimization, and compute scalability simultaneously for fine-tuning (in particular)? *Frame Theory* provides a natural answer. Briefly, Finite Frames and Fusion Frames are used to study the spanning sets and spanning subspaces of a given finite-dimensional vector space, respectively. They have many applications in coding theory, compressed sensing (Boufounos et al., 2009), quantization (Adepu et al., 2024; Czaja & Na, 2024), and dictionary learning (Hwang et al., 2019; Cai et al., 2014). Their ability to encode updates in overlapping subspaces offers robustness, flexibility, and efficiency – properties that are relevant for model adaptation. FrameFT partitions the model's parameter space into fusion-frame-based subspaces, within which updates are learned in a highly sparse and structured manner.

**Our proposal: Fine-tuning with Frames.** We propose a new PEFT strategy inspired by Fusion Frames (Christensen, 2018; Casazza et al., 2008; Waldron, 2019). Fusion Frames allow structured representations by decomposing a space into overlapping subspaces at multiple scales. Unlike strategies that rely on either low-rank or sparse updates, FrameFT performs fine-tuning across multiple structured subspaces, naturally combining the global adaptability of LoRA with the precision of sparse updates.

**Contributions.** Our main contributions are: **(a)** A new fine-tuning framework, **FrameFT**, that utilizes highly structured subspaces (Fusion Frames) to capture parameter updates efficiently; **(b)** Extensive empirical evaluations across multiple foundation models and tasks, demonstrating FrameFT's effectiveness compared to state-of-the-art PEFT baselines; **(c)** A technical analysis of the numerical stability, convergence, and efficiency benefits offered by FrameFT.

**Conflict of Interest Disclosure.** Two of the authors (LZ, SK) work at Google DeepMind, where the Gemma models were developed. The other two authors (HA, VS) previously worked at Google DeepMind.

## 2. Frames and Subspace decompositions

We provide a brief review of Finite Frame theory for Hilbert spaces (Waldron, 2019; Christensen, 2018) . Readers familiar with these concepts can skip to the next section, where we describe how we apply this idea to fine-tuning.

**Spanning sets and Frames.** A spanning set is a collection of vectors that spans a finite-dimensional Hilbert space or vector space, and orthonormal bases are a canonical example. Frames generalize this concept by allowing redundancy,

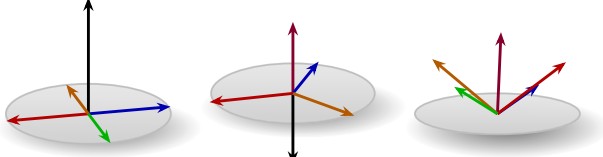

*Figure 1.* Examples of tight frames with $k = 5$ vectors in $\mathbb{R}^3$.

enabling stable and often more robust representations.

Let $\mathcal{H}^d$ denote a $d$-dimensional Hilbert space. A sequence of vectors $\phi = (\varphi_i)_{i=1}^k$ in $\mathcal{H}^d$ is called a frame if there exist constants $0 < A \leq B < \infty$ such that for all $x \in \mathcal{H}^d$,

$$A||x||^2 \leq \sum_{i=1}^k |\langle x, \varphi_i \rangle|^2 \leq B||x||^2, \tag{1}$$

where $\langle \cdot, \cdot \rangle$ is the inner product in $\mathcal{H}^d$, and $A$ and $B$ are known as the frame bounds. Typically, $k \geq d$ allows the frame to represent vectors with redundancy. An example is illustrated in Figure 1 with $k = 5$ vectors in $\mathbb{R}^3$. We can project any vector in $\mathbb{R}^3$ onto these frame vectors without distorting the signal (Casazza & Kutyniok, 2012). Throughout the paper, we operate in real-valued Euclidean spaces $\mathbb{R}^d$ instead of a more general Hilbert space for simplicity.

Fusion frames extend the above idea to settings where subspace decomposition is also desired (instead of redundancy alone). Given a collection of subspaces $\{\mathcal{W}_i\}_{i=1}^k$ in $\mathbb{R}^d$ and a corresponding set of positive weights $\{w_i\}_{i=1}^k$, the collection $(\mathcal{W}_i, w_i)_{i=1}^k$ forms a fusion frame if there exist constants $0 < A \leq B < \infty$ such that for all $x \in \mathbb{R}^d$,

$$A||x||^2 \leq \sum_{i=1}^k w_i^2 ||P_i x||^2 \leq B||x||^2,$$

where $P_i$ denotes the orthogonal projection onto the subspace $\mathcal{W}_i$. The weights $w_i$ adjust the influence of each subspace, and $A$, $B$ are fusion frame bounds. A fusion frame is tight if $A = B$, and Parseval if $A = B = 1$. We will focus on Parseval fusion frames. If all weights $w_i$ are equal to 1, we write the fusion frame simply as $\{\mathcal{W}_i\}_{i=1}^k$. Frame bounds will help later in our convergence analysis.

### 2.1. Operators in Fusion Frames

Let $((\mathcal{W}_i, w_i))_{i=1}^k$ be a Fusion Frame for $\mathbb{R}^d$, with orthogonal projection matrices $(P_i)_{i=1}^k$ respectively. We define three operators:

The analysis operator $\mathcal{T}_\mathcal{W} : \mathbb{R}^d \to \bigoplus_{i=1}^k \mathcal{W}_i$ maps a vector to its projections across all subspaces:

$$\mathcal{T}_\mathcal{W}(x) = (w_i P_i^T x)_{i=1}^k. \tag{2}$$

The synthesis operator $\mathcal{T}_\mathcal{W}^* : \bigoplus_{i=1}^k \mathcal{W}_i \to \mathbb{R}^d$ "re-builds" a

vector from its fusion frame representation:

$$\mathcal{T}_{\mathcal{W}}^*((y_i)_{i=1}^k) = \sum_{i=1}^k w_i P_i y_i. \tag{3}$$

The fusion frame operator is the composition of the two operators above, $\mathcal{S}_{\mathcal{W}} = \mathcal{T}_{\mathcal{W}}^* \mathcal{T}_{\mathcal{W}}$, defined by:

$$\mathcal{S}_{\mathcal{W}}(x) = \sum_{i=1}^k w_i^2 P_i P_i^T x. \tag{4}$$

This operator is self-adjoint, positive semi-definite, and bounded. For Parseval fusion frames, we have $\mathcal{S}_{\mathcal{W}} = I_d$ (allow exact reconstruction). These operators help in mapping vectors in $\mathbb{R}^d$ to their fusion frame representations and back.

## 3. Fine-tuning in Fusion Frame Subspaces

We will start by analyzing the subspace updates in LoRA (Hu et al., 2022). The key idea is representing weight updates through a low-rank decomposition. For a layer $l$ with pre-trained parameters $W_l \in \mathbb{R}^{m \times n}$, LoRA decomposes the weight update $\delta W_l$ as a product of two rank-deficient matrices:

$$W_l' = W_l + \delta W_l = W_l + B_l A_l \tag{5}$$

where $B_l \in \mathbb{R}^{m \times r}$, $A_l \in \mathbb{R}^{r \times n}$, and $r < \min(m, n)$.

To better understand the structure of these updates, we can check the Singular Value Decomposition (SVD) of $\delta W_l$:

$$\delta W_l = U \Sigma V^T = U_r \Sigma_r V_r^T \tag{6}$$

where $U \in \mathbb{R}^{m \times m}$ and $V \in \mathbb{R}^{n \times n}$ are best viewed as orthogonal matrices spanning the *output* and *input* spaces respectively, and $\Sigma \in \mathbb{R}^{m \times n}$ is a rectangular diagonal matrix. Since the rank of $\delta W_l$ is constrained to $r$, we can simplify this product to $U_r \Sigma_r V_r^T$, where $\Sigma_r$ contains only the top $r$ singular values and $U_r, V_r$ are the corresponding left and right singular vectors. This decomposition provides insight into LoRA updates: $V_r$ defines a subspace in the *input space* $\mathbb{R}^n$ where the input is first projected. Then, $\Sigma_r$ scales these projections, and $U_r$ maps them to a subspace of the *output space* $\mathbb{R}^m$. These input and output subspaces emerge *implicitly* as a by-product of training the matrices $B$ and $A$, without direct control over their properties or interactions.

**Asymmetricity.** Recent results indicate that the contribution of matrices $A$ and $B$ is *asymmetric*, which can lead to instability (Hayou et al., 2024), or failure to converge to the optimal solution (Malinovsky et al., 2024). This suggests two key opportunities for improvement. **(i)** Can we explicitly choose and work with multiple subspaces in both input

space $\mathbb{R}^n$ and output space $\mathbb{R}^m$ to better capture parameter updates? **(ii)** Can we maintain precise control over how these subspaces interact during training?

While existing work has explored defining subspaces using Orthogonal bases derived from the pretrained weights (Lingam et al., 2024; Sun et al., 2024a) or fixed spectral bases such as the Fourier basis (Gao et al., 2024), there is a significant cost in storage and impact on inference throughput due to dense operations. We will see how Fusion frames address **both questions** directly. Specifically, we can decompose the input and output spaces into potentially overlapping subspaces with controlled relationships. Further, we can achieve good storage efficiency and inference throughput due to the inherent sparsity of the generated frames for free.

### 3.1. Fusion Frames for Finetuning Weight Updates

For the first problem, we want to decompose the input and output spaces into multiple subspaces. The Fusion Frame operators in §2.1 can be used. In particular, let $(P_{n,i}^T)_{i=1}^k$ and $(P_{m,i}^T)_{i=1}^k$ be the orthogonal projection matrices for decomposing the input ($\mathbb{R}^n$) and output ($\mathbb{R}^m$) spaces into $k$ subspaces with dimensions $\rho_n$ and $\rho_m$, respectively. Let,

$$\begin{aligned} P_n &= [P_{n,1}, P_{n,2}, \dots, P_{n,k}] \\ P_m &= [P_{m,1}, P_{m,2}, \dots, P_{m,k}] \end{aligned} \tag{7}$$

We can model rich interactions between these subspaces by using trainable coefficient matrices, in contrast to the diagonal matrix $\Sigma_r$ in LoRA. To be specific, we define $(C_{l,i})_{i=1}^k$ where each $C_{l,i} \in \mathbb{R}^{\rho_m \times \rho_n}$ encodes the relationships within the $i$-th input and output subspaces. These matrices can be neatly organized into a block diagonal structure: $C_l = \mathrm{diag}(C_{l,1}, C_{l,2}, \dots, C_{l,k})$ which yields our simple parameter update equation:

$$W_l' = W_l + \delta W_l = W_l + \frac{\alpha}{\sqrt{mn}} P_m C_l P_n^T$$

Expanding this expression with the subspaces, we get:

$$W_l' = W_l + \frac{\alpha}{\sqrt{mn}} \sum_{i=1}^k P_{m,i} C_{l,i} P_{n,i}^T$$

Here, $\alpha$ is a scaling hyperparameter, normalized by $\sqrt{mn}$ (for dimension independence). Fig. 2 (*left*) shows the difference between SVD in LoRA and FrameFT. The projection matrices $P_n$ and $P_m$ are obtained from Tight Fusion Frames (TFF) discussed in §3.2. Under certain conditions, these TFFs exhibit Equichordal and Equi-Isoclinic properties (Fickus et al., 2023b), maximizing subspace *separation* for efficient representations. These TFFs remain **fixed** and **only the coefficient matrices** $(C_{l,i})_{i=1}^k$ are trainable. Fig. 2 (*right*) shows FrameFT update for a single layer (also see Alg. 1).

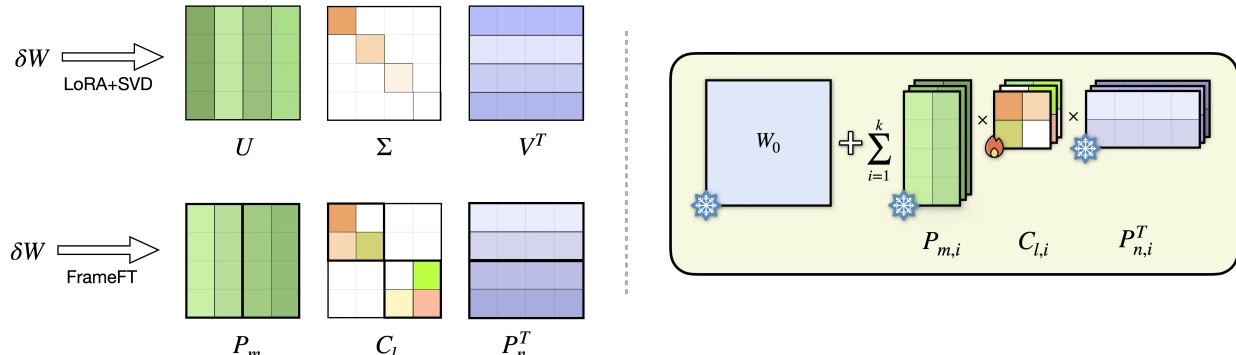

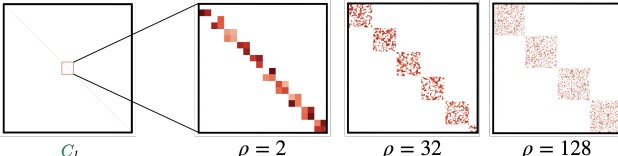

*Figure 2.* (*left*) How parameter update $\delta W$ is modeled by LoRA and by FrameFT. (*right*) The parameter update rule in FrameFT. We freeze the Fusion Frame projection matrices and train only the coefficients $C_l$.

---

**Algorithm 1** FrameFT: Finetuning with Frames

---

**Require:** Frozen Parameters $W_l$, Fusion Frame number of subspaces $k$, subspace dimensions $\rho_m, \rho_n$, number of non-zero coefficients $n_c$, non-zero coefficients $G$, scalar $\alpha$

1: $m,n = \text{shape}(W_l)$     // get the shape of the parameter matrix
2: $P_m, P_n = \text{TFF}(k, \rho_m, m), \text{TFF}(k, \rho_n, n)$     // generate the TFFs
3: $\text{Coeffs} = \text{Init}(m, n, \text{size}=n_c)$     // initialize the coefficients matrix
4: $C_l = \text{rearrange}(\text{Coeffs}, m, n, G)$     // re-arrange the coefficients in dense format
5: $\delta W_l = \frac{\alpha}{\sqrt{mn}} P_m C_l P_n^T$     //compute the update term
6: $W_l^{'} = W_l + \delta W_l$     // update the parameter matrix

---

### 3.2. Constructing Tight Fusion Frames

We use uniform tight fusion frames, where all subspaces have the *same* dimension. In practice, one can easily use different subspace dimensions across layers. To construct a $(k, \rho, d)$ uniform tight fusion frame where $k$ is the number of subspaces, $\rho$ is the dimension of each subspace, and $d$ is the dimension of the entire space, we use the Spectral Tetris algorithm described in (Casazza et al., 2011), which is summarized as: (1) Construct a unit norm tight frame (UNTF) in $\mathbb{C}^\rho$ with $d$ vectors. (2) Modulate these vectors using the $k^{\text{th}}$ roots of unity to form $k$ subspaces of dimension $\rho$ in $\mathbb{C}^d$. (3) Use the method in (Fickus et al., 2023a) to extend the result to real-valued spaces. This process ensures that the resultant set satisfies the tight fusion frame conditions. An example is included in Appendix §H.

### 3.3. Structured Coefficient Matrix

Prior work on Sparse Fine-Tuning methods (Ansell et al., 2022) suggests that we can boost parameter efficiency by introducing sparsity into the coefficient matrices $(C_{l,i})_{i=1}^k$. This sparsification nicely complements our fusion frame architecture: while fusion frames provide *structured subspace decompositions of the parameter space*, sparse coefficients identify the *most useful subspace interactions* (see off-diagonal terms in Fig. 2). Note that the Fusion

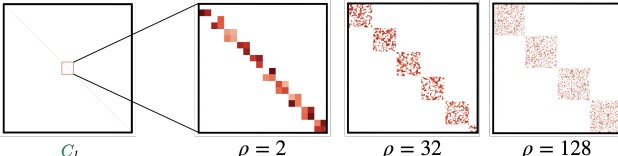

*Figure 3.* An example of the coefficient matrix $C_l$ from a Query layer (RoBERTa-L) with different subspace dimensions $\rho_m = \rho_n = \rho$.

Frames themselves are sparse, which is *distinct* from the sparsity in the coefficient matrix. Both are exploited in our implementation. Our experiments indicate that a simple random sparsity pattern performs well: we randomly designate a small subset of entries in $C_l$ as non-zero, distributing them uniformly across the block-diagonal matrices $(C_{l,i})_{i=1}^k$. Figure 3 shows this sparse structure with an example from RoBERTa-Large. The memory requirements for storing the updates reduce from $O(k\rho_m\rho_n)$ to $O(s)$, where $s \ll k\rho_m\rho_n$. To reduce memory overhead even further, we share these sparsity patterns across layers, i.e., $\text{supp}(C_l) = \text{supp}(C_{l'})$ for layers $l$ and $l'$, where $\text{supp}(\cdot)$ represents the sets of index locations. The cross-layer sharing of sparsity patterns not only reduces memory requirements but also suggests that subspace interactions necessary for task adaptation may share structural patterns across network layers. An ablation study we present in §4.6 confirms this empirically: shared and per-layer random masks perform similarly (plus/minus noise of each other).

### 3.4. System Efficiency and Scalability

Our design choices, specifically the use of algorithmically generated Fusion Frames and sparse coefficients, give a number of advantages over alternatives.

**Storage efficiency.** Since Spectral Tetris generates Fusion Frames algorithmically, we eliminate the need to store basis matrices. Only the sparse coefficient matrices must be serialized. For a Llama-2-7B model, this reduces the checkpoint size to just 1.28 MB (vs. 67.1 MB for LoRA rank-32), facilitating ultra-low-latency "hot-swapping" of

adapters in multi-tenant serving scenarios where a single backbone supports thousands of fine-tuned models for different users. Both numbers above assume FP32 precision. The frozen backbone operates in FP16/BF16 during training, but adapter parameters can be maintained in FP32 to preserve dynamic range/gradient fidelity.

**Memory Overhead.** Unlike SVD-based methods such as SVFit (Sun et al., 2024a) and LoRA-XS (Bałazy et al., 2025), which require unique basis vectors for each layer, Fusion Frames generated by the Spectral Tetris algorithm depend *only* on the layer dimensions. This allows a single set of frames to be **shared across all layers** with the same dimensions, significantly reducing the initialization time (see Table 11) and runtime memory footprint.

**Inference throughput.** While the orthogonal bases generated by SVD based methods are dense, Spectral Tetris directly constructs frames with inherent block-sparsity (see §10). This structural advantage directly translates to higher inference throughput (see Table 2).

Beyond these practical efficiencies, the construction of FrameFT also offers useful theoretical advantages for optimization. Next, we analyze how preserving the smoothness of the loss landscape allows stable convergence properties.

### 3.5. Analysis of FrameFT Convergence properties

Our formulation of the parameter updates using Tight Fusion Frames and sparse coefficients enjoys theoretical benefits compared to LoRA. The analysis in Sun et al. (2024b) describes how, for Lipschitz-smooth loss functions, LoRA can create a non-smooth landscape when projected onto the parameters in $A$ and $B$. Via Lemma 3.1, we show that FrameFT preserves Lipschitz-smoothness: when a function exhibits Lipschitz smoothness w.r.t. $W$, this property holds when remapped onto our coefficient space $C_l$.

**Lemma 3.1.** *For any differentiable function $f(W)$ that is $L$-Lipschitz smooth under Frobenius norm, with fusion frames characterized by frame bounds $(A_m, B_m)$ and $(A_n, B_n)$ for output and input spaces, respectively, the transformed function $f(C_l) = f(W_0 + P_m C_l P_n^T)$ obeys:*

$$||\nabla f(C_l^1) - \nabla f(C_l^2)||_F \leq L B_m B_n ||C_l^1 - C_l^2||_F$$

Note that $f(W)$ denotes the global loss function of the entire deep neural network, defined over the collection of all layer weights $W = \{W_1, \ldots, W_D\}$ just like in (Sun et al., 2024b, see p. 13). Thus, $f$ implicitly encodes the composition of all network layers. During fine-tuning, with pre-trained weights frozen, the loss landscape is reparameterized purely as a function of the coefficient matrices $f(C)$, where $C = \{C_1, \ldots, C_D\}$ represents the set of learnable Frame coefficients across all layers.

Since FrameFT preserves smoothness, we can immediately invoke a broad set of convergence results for Lipschitz-smooth functions (Bubeck, 2015; Zhou & Cong, 2017). We highlight one result: gradient descent constrained to learning rates $\eta \leq 1/\tilde{L}$ achieves the expected $1/T$ convergence rate to first-order stationary points (in Theorem 3.2 below). Additional details are provided in §B.

**Theorem 3.2.** *Consider minimizing $f(C_l) = f(W_0 + P_m C_l P_n^T)$ where: $f(C_l)$ is $\tilde{L}$-Lipschitz smooth (but potentially non-convex) and $f(C_l)$ is lower bounded by $f^*$. For gradient descent with step size $\eta \leq 1/\tilde{L}$, running gradient descent for $T$ iterations satisfies:*

$$(1/T) \sum_{t=0}^{T-1} ||\nabla f(C_l^t)||_F^2 \leq 2(f(C_l^0) - f^*)/(\eta T)$$

This analysis offers a unified view of subspace-based fine-tuning. Our results can be extended to methods such as SVFit and FourierFT by substituting the appropriate projection matrices and bounds. Lemma 3.1 explicitly links the Lipschitz constant of the reparameterized loss $f(C)$ to the upper frame bounds $B_m, B_n$. This result directly guides our design choice: to maximize the smoothness of the loss landscape, we normalize our Frame bounds to unity (Parseval frames) for all experiments. To summarize, Lemma 3.1 informs the design choices via the bound $\tilde{L} = L B_m B_n$.

## 4. Experiments

In this section, we compare the performance of Transformer-based models fine-tuned with FrameFT. All our experiments are conducted on two NVIDIA A100 GPUs with 40GB of memory each. Individual subsections include more details on the experimental setup. Additional experiments on varying hyperparameters and discussions on subspace dimensions are available in the Appendix §D, §E, and §F.

### 4.1. Natural Language Understanding Capabilities

**Evaluation framework:** We evaluate the performance of FrameFT by fine-tuning the base and large variants of RoBERTa (Liu et al., 2019) across multiple GLUE benchmark tasks (Wang et al., 2019). This suite of benchmarks covers sentiment classification, paraphrase detection, and entailment recognition. It is a standardized setup for testing.

**Fine-tuning strategy:** We use the original LoRA (Hu et al., 2022) recipe by fine-tuning the Query and Value matrices across network layers. For FrameFT, we use 1000 non-zero coefficient entries with their positions determined randomly and shared across layers. For tight fusion frame construction, we use a subspace dimension $\rho = 2$, and the subspace count $k$ is calibrated so that $k\rho = n$ ($n$ is layer dimension).

*Table 1.* Fine-Tuning RoBERTa Base and Large models on GLUE benchmark. * indicates the results reported in prior work. FrameFT performs better than full fine-tuning and LoRA, using $10\times$ fewer parameters.

| Model | Method | Params | SST-2 | MRPC | CoLA | QNLI | RTE | STS-B | Avg. |
|---|---|---|---|---|---|---|---|---|---|
| RoBERTa Base | FF* | 125M | 94.8 | 90.2 | 63.6 | **92.8** | 78.7 | 91.2 | 85.2 |
| | SMT | 55K | 94.2 | 89.5 | 63.9 | 91.8 | 78.1 | 90.4 | 84.7 |
| | LoRA* | 0.3M | **95.1** | 89.7 | 63.4 | 93.3 | 78.4 | **91.5** | 85.2 |
| | SVFT$_{d=2}^{R}$ | 92K | 94.3 | 89.4 | 62.4 | 92.0 | 78.3 | 91.1 | 84.6 |
| | AdaLoRA* | 0.3M | 94.5 | 88.7 | 62.0 | 93.1 | **81.0** | 90.5 | 85.0 |
| | FourierFT* | 24K | 94.2 | 90.0 | 63.8 | 92.2 | 79.1 | 90.8 | 85.0 |
| | SVFit* | 18K | 92.4 | 90.0 | 63.8 | 90.8 | 78.0 | 92.4 | 85.1 |
| | FrameFT | 24K | 94.3 | **92.3** | **66.8** | 92.4 | 79.8 | 90.9 | **86.1** |
| RoBERTa Large | FF* | 356M | **96.4** | 90.9 | 68.0 | 94.7 | 86.6 | **92.4** | 88.2 |
| | SMT | 1.5M | 96.0 | 89.7 | 69.4 | 93.6 | 84.1 | 91.8 | 87.4 |
| | LoRA* | 0.8M | 96.2 | 90.2 | 68.2 | **94.8** | 85.2 | 92.3 | 88.2 |
| | SVFT$_{d=2}^{R}$ | 0.25M | 96.1 | 90.2 | 66.7 | 94.3 | 83.0 | 92.1 | 87.1 |
| | VeRA | 61K | 96.1 | 90.9 | 68.0 | 94.4 | 85.9 | 91.7 | 87.8 |
| | LoRA-XS* | 60K | 96.3 | 91.2 | 68.6 | 94.3 | **89.5** | 92.2 | **88.7** |
| | RoseLoRA* | 53.4K | 95.2 | 90.2 | 69.2 | 94.7 | 89.2 | 92.0 | 88.5 |
| | FourierFT* | 48K | 96.0 | 90.9 | 67.1 | 94.4 | 87.4 | 91.9 | 88.0 |
| | SVFit* | 36K | 96.2 | 90.9 | **71.4** | 94.4 | 86.3 | 92.0 | 88.5 |
| | FrameFT | 48K | 96.2 | **92.6** | 69.8 | 93.4 | **88.1** | 91.9 | **88.7** |

**Performance analysis:** We report the Pearson correlation coefficient (PCC) for the STS-B task, Matthews correlation coefficient (MCC) for CoLA and accuracy for the remaining tasks. For the baseline methods, we report LoRA (Hu et al., 2022), AdaLoRA (Zhang et al., 2023), FourierFT (Gao et al., 2024), SVFit (Sun et al., 2024a), SMT (He et al., 2025), SVFT (Lingam et al., 2024) and RoseLoRA (Wang et al., 2024). The results are presented in Table 1. We see that FrameFT performs on par or better than LoRA and full fine-tuning on individual tasks, and performs better than all the baselines on average. FrameFT achieves this with $10\times$ fewer parameters when compared to LoRA. We also note that even though the number of parameters of SVFit (Sun et al., 2024a) is slightly lower than FrameFT, SVFit trains the singular values, keeping the singular vectors fixed. So, in practice, one would need to save the singular vectors for each layer after training, which increases the storage requirements. Also, for RoseLoRA (Wang et al., 2024), only the parameter count is reported in Table 1, but they use a different mask for each layer, so the storage cost is $3\times$ the number of parameters if we account for the location of the non-zero coefficients.

### 4.2. Instruction Tuning

**Benchmarking Framework:** We evaluate FrameFT for fine-tuning LLMs to follow instructions. We finetune 7B and 13B variants of Llama2 models (Touvron et al., 2023), Gemma2 models (Team et al., 2024b) with 2B and 9B parameters and Llama 3.1 (Grattafiori et al., 2024) 8B model on the Alpaca instruction dataset (Taori et al., 2023). We evaluate the performance of the fine-tuned models on the LM-evaluation harness (Gao et al., 2023) by Eleuther AI. We use eight distinct challenge categories spanning reason-

ing, world knowledge, and generalization capabilities.

**FrameFT configuration:** We apply FrameFT with $n = 5000$ non-zero coefficients and adapt the Query and Value matrices for all the Transformer blocks. We determine the position of the non-zero coefficients at random and share them across all the layers. We use $\alpha = 200$ across all of our experiments. A sensitivity analysis sweeping $\alpha \in \{10, 100, 400, 600\}$ for Llama-2-7B is in the Appendix I. We set the subspace dimension $\rho = 2$ and as before, calculate the number of subspaces $k$ such that $k\rho = n$ for each layer. For all baselines, we choose the hyperparameters suggested for the respective method.

**Performance Analysis:** We compare FrameFT with LoRA (Hu et al., 2022), $(IA)^3$ (Liu et al., 2022a), DoRA (Liu et al., 2024), $S^2FT$ (Yang et al., 2024), SVFT (Lingam et al., 2024), and FourierFT (Gao et al., 2024). Our results are shown in Table 3. We observe that FrameFT performs on par or better compared to the baselines while using the fewest number of parameters across all models. In addition, FrameFT also provides other compute benefits we discuss in Section 4.4.

### 4.3. Performance on Vision transformer models

We performed a set of experiments to determine whether FrameFT was effective only for Language models. To check this, we applied FrameFT to fine-tune Vision Transformers on 8 image classification tasks, which include remote sensing, fine-grained classification, and texture recognition.

**Performance Analysis:** Section C in the appendix shows the performance of FrameFT across these tasks for ViT-L and ViT-B models. We observe that FrameFT performs better than the baseline methods (which include full-finetuning) on average. Moreover, we observe performance improvements across the majority of the tasks when we increase the number of parameters for FrameFT. These results indicate that FrameFT generalizes well across both Vision and Language models. More experimental details are presented in Appendix C and Table 8.

*Table 2.* Tokens per second for different PEFT methods.

| | Method | #params | #toks/sec |
|---|---|---|---|
| Llama-2-7b | LoRA | 262k | 23.6k |
| | DoRA | 262k | 14.1k |
| | SVFit | 1k | 9.0k |
| | FourierFT | 1k | 1.8k |
| | FrameFT | 1k | **39.4k** |
| Gemma-2-9b | LoRA | 245k | 22.1k |
| | DoRA | 245k | 16.0k |
| | SVFit | 1k | 12.4 |
| | FourierFT | 1k | 1.4k |
| | FrameFT | 1k | **29.6k** |
| Llama-3.1-8b | LoRA | 163k | 22.3k |
| | DoRA | 163k | 18.7k |
| | SVFit | 1k | 11.5k |
| | FourierFT | 1k | 1.6k |
| | FrameFT | 1k | **27.3k** |

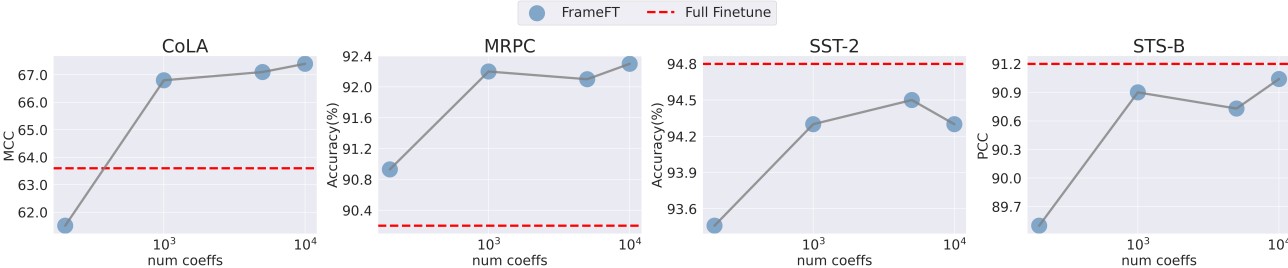

*Figure 4.* Performance of RoBERTa base model fine-tuned with FrameFT versus the number of non-zero coefficients. The red line indicates the performance of full fine-tuning.

### 4.4. Operational Efficiency/Latency Analysis

In this section, we analyze the operational efficiency of FrameFT in terms of storage requirements and inference throughput. In practice, when serving LLMs to a large user base, the pretrained model is held fixed, and various specialized variants are maintained simultaneously. Hence, we measure the throughput of the adapter layer introduced by different PEFT methods, with the idea that the pretrained layer throughput remains the same in the pretrained and fine-tuned models.

**Inference throughput:** Table 2 displays the tokens per second for various methods across different model families. All measurements are performed on an NVIDIA A100 machine. We observe that FrameFT performs better than LoRA, which in turn performs better than FourierFT and other methods. These measurements are near-deterministic on fixed hardware. Across 100 iterations the coefficient of variation is below 12%, and the FrameFT–LoRA gap is large enough that variance does not affect the conclusion (e.g., on Llama-2-7B: $37.9 \pm 4.5$k for FrameFT vs. $24.4 \pm 2.7$k tokens/sec for LoRA). This highlights FrameFT's efficiency benefits due to sparse coefficients and sparse projection matrices as described in §3.3 and Appendix §H

**Storage and Memory Footprint.** A key advantage of FrameFT is its minimal storage footprint for task-specific adapter checkpoints. Since Fusion Frames are generated algorithmically once and shared across layers of the same dimension, the basis vectors $(P_m, P_n)$ do not need to be stored or transmitted. We only store the sparse coefficient matrices. This is a clear advantage compared to SVD-based methods such as SVFT and LoRA-XS, where one needs to store the dense orthogonal basis for each layer, increasing their storage/memory requirements.

We note that constructing the Fusion Frames takes a fixed amount of time. Specifically, the time complexity for constructing a $(k, \rho, d)$ TFF is $\mathcal{O}(kd)$. However, this generation step is performed only **once** during initialization, and the resulting projection matrices are shared across all network layers, amortizing the one-time compute cost across multi-

ple layers and forward calls (see §H.3 for wall clock time). So, we focus on per-layer throughput measurements.

### 4.5. Performance vis-a-vis non-zero coefficients

We evaluate the performance of FrameFT as we increase the number of non-zero coefficients. We choose the RoBERTa base model for this experiment and check the performance of the model fine-tuned with FrameFT on different tasks in the GLUE benchmark. Figure 4 shows expected trends: the performance of the fine-tuned model on different tasks increases as we increase the number of non-zero terms in the coefficient matrix. Table 6 also confirms this behavior as we increase $n$ from 1000 to 5000. These results show that FrameFT is able to utilize the additional parameters to improve the fine-tuning performance for both vision and language tasks. Additional experiments on more tasks from the GLUE benchmark are included in appendix E.

### 4.6. Sparsity Pattern and Cross-Layer Sharing Ablation

In this experiment, we ablate the choice of random shared sparsity by comparing four masking strategies – a random shared mask across all layers, a per layer random mask, magnitude-based non-zero coefficients selection and a gradient-based mask for FrameFT. We evaluate the performance of each of these methods for finetuning RoBERTA-base model on the GLUE benchmark. For the random mask strategies, we average the performance accross five different seeds each. Results are shown in Table 4.

We see that cross-layer sharing incurs no expressivity cost: shared and per-layer masks are indistinguishable within variance. This aligns with Lemma 3.1: the optimization geometry depends on the Fusion Frame bounds and not on the specific entries of the coefficient matrix, which are non-zero. Interestingly, adaptive heuristics perform strictly worse. Gradient-based masks collapse to 74.83 average, lower than random. We suspect this is due to task-relevant subspace interactions being broadly distributed across the coefficient matrix rather than concentrated in high-magnitude or high-gradient pretrained weights. Random sampling provides better coverage of this distributed structure than greedy se-

*Table 3.* Performance of LLMs fine-tuned on the Alpaca dataset by various methods and then evaluated on the LM-evaluation-harness. FrameFT performs competitively with all the baseline methods under comparison, using a $30\times$ fewer number of parameters than LoRA. FrameFT needs minimal hyperparameter tuning.

| Model | Method | #Params | ARC-c | ARC-e | BoolQ | HellaSwag | OBQA | PIQA | RTE | WinoGrande | Avg. |
|---|---|---|---|---|---|---|---|---|---|---|---|
| Llama-2-7b | LoRA | 16.7M | 45.82 | 77.02 | 78.81 | 58.08 | 35.2 | 78.83 | 61.01 | 70.72 | 63.18 |
| | $(IA)^3$ | 614K | 45.38 | 76.60 | 77.89 | 57.74 | 34.6 | 78.78 | 64.26 | 68.82 | 63.01 |
| | DoRA | 16.7M | 45.14 | 76.39 | 78.35 | 57.78 | 34.0 | 78.73 | 67.15 | 67.17 | 63.09 |
| | $S^2FT$ | 56.6M | 46.16 | 77.44 | 78.38 | 58.10 | 32.8 | 78.84 | 61.01 | 68.58 | 62.66 |
| | $SVFT^B_{d=8}$ | 4.5M | 45.79 | 77.07 | 78.83 | 57.98 | 34.6 | 78.56 | 61.52 | 70.64 | 63.12 |
| | FourierFT | 320K | 44.96 | 77.14 | 79.05 | 58.21 | 34.6 | 78.89 | 62.45 | 70.48 | 63.22 |
| | FrameFT | 320K | 45.22 | 76.93 | 78.62 | 58.08 | 34.2 | 78.62 | 66.06 | 71.19 | **63.62** |
| Llama-2-13b | LoRA | 26.2M | 50.77 | 80.35 | 81.44 | 61.13 | 36.0 | 79.38 | 69.68 | 72.61 | 66.42 |
| | $(IA)^3$ | 963K | 50.25 | 79.96 | 80.52 | 60.54 | 34.2 | 79.27 | 68.59 | 72.22 | 65.69 |
| | DoRA | 26.2M | 51.79 | 80.13 | 80.21 | 61.44 | 35.6 | 79.81 | 71.48 | 72.21 | **66.58** |
| | $S^2FT$ | 111M | 20.73 | 35.10 | 46.17 | 32.94 | 14.6 | 58.48 | 55.23 | 50.04 | 39.16 |
| | $SVFT^B_{d=8}$ | 7.1M | 50.45 | 80.42 | 80.51 | 60.82 | 34.8 | 79.22 | 69.58 | 72.06 | 65.98 |
| | FourierFT | 400K | 49.74 | 80.17 | 80.82 | 60.98 | 35.6 | 79.76 | 64.62 | 72.06 | 65.47 |
| | FrameFT | 400K | 50.34 | 79.75 | 81.19 | 60.87 | 35.8 | 80.08 | 68.59 | 72.29 | 66.11 |
| Gemma-2-2B | LoRA | 6.4M | 48.63 | 79.76 | 76.61 | 55.85 | 31.6 | 78.94 | 62.09 | 68.19 | 62.70 |
| | $(IA)^3$ | 292K | 46.76 | 80.13 | 70.70 | 55.79 | 31.2 | 78.13 | 59.21 | 69.77 | 61.46 |
| | DoRA | 6.4M | 48.72 | 80.35 | 72.26 | 55.96 | 33.6 | 78.56 | 67.15 | 68.98 | 63.20 |
| | $S^2FT$ | 16.9M | 45.90 | 78.41 | 59.24 | 54.80 | 31.4 | 77.80 | 57.40 | 56.27 | 57.65 |
| | $SVFT^B_{d=8}$ | 1.4M | 48.72 | 81.10 | 71.13 | 56.37 | 33.6 | 78.84 | 65.70 | 70.40 | 63.23 |
| | FourierFT | 260K | 45.98 | 79.12 | 73.88 | 55.57 | 32.8 | 79.16 | 67.87 | 67.87 | 62.78 |
| | FrameFT | 260K | 48.37 | 81.06 | 75.13 | 55.71 | 34.6 | 79.05 | 69.67 | 69.30 | **64.11** |
| Gemma-2-9B | LoRA | 17.9M | 64.76 | 88.22 | 86.39 | 62.96 | 36.2 | 82.48 | 70.40 | 75.30 | **70.83** |
| | $(IA)^3$ | 774K | 61.95 | 87.21 | 85.05 | 62.11 | 35.8 | 81.56 | 68.95 | 74.27 | 69.61 |
| | DoRA | 17.9M | 62.62 | 87.16 | 86.02 | 62.91 | 35.2 | 81.61 | 71.48 | 73.71 | 69.09 |
| | $S^2FT$ | 74.4M | 53.41 | 80.55 | 80.73 | 59.02 | 32.8 | 79.54 | 69.67 | 70.32 | 65.76 |
| | $SVFT^B_{d=8}$ | 3.9M | 63.25 | 87.88 | 86.54 | 63.11 | 36.2 | 81.72 | 71.12 | 73.64 | 70.43 |
| | FourierFT | 420K | 64.16 | 88.21 | 86.36 | 62.78 | 36.4 | 81.66 | 70.03 | 74.27 | 70.48 |
| | FrameFT | 420K | 65.01 | 88.00 | 86.02 | 63.09 | 37.2 | 81.72 | 69.31 | 74.82 | 70.65 |
| Llama-3.1-8B | LoRA | 13.6M | 55.38 | 83.38 | 82.08 | 61.73 | 35.2 | 81.23 | 74.01 | 75.53 | **68.56** |
| | $(IA)^3$ | 524K | 54.27 | 82.79 | 82.20 | 61.04 | 34.4 | 80.58 | 68.95 | 74.59 | 67.35 |
| | DoRA | 13.6M | 54.01 | 82.07 | 81.31 | 61.07 | 34.6 | 81.23 | 67.51 | 72.77 | 66.82 |
| | $S^2FT$ | 65.4M | 31.57 | 60.02 | 59.82 | 44.24 | 20.0 | 68.88 | 56.68 | 53.67 | 49.36 |
| | $SVFT^B_{d=8}$ | 2.8M | 53.12 | 82.39 | 82.32 | 60.58 | 36.2 | 80.53 | 73.36 | 74.02 | 67.81 |
| | FourierFT | 320K | 51.36 | 80.51 | 81.49 | 60.57 | 34.0 | 80.41 | 70.39 | 73.95 | 66.58 |
| | FrameFT | 320K | 53.41 | 82.53 | 82.29 | 60.79 | 34.6 | 80.90 | 75.09 | 73.40 | 67.87 |

*Table 4.* Sparsity mask ablation on RoBERTa-Base/GLUE (averaged over five seeds). Shared and per-layer random masks perform within noise of each other while adaptive heuristics perform strictly worse.

| Method | SST-2 | MRPC | CoLA | QNLI | RTE | STS-B | Avg. |
|---|---|---|---|---|---|---|---|
| Shared random mask | 94.40±0.17 | 92.31±0.46 | 66.58±0.77 | 92.35±0.12 | 80.21±0.82 | 90.96±0.11 | 86.18±0.18 |
| Per-layer random mask | 94.10±0.16 | 93.53±0.32 | 66.44±0.61 | 92.31±0.14 | 80.34±0.39 | 90.86±0.09 | 86.26±0.17 |
| Magnitude-based mask | 94.19 | 92.49 | 65.82 | 91.61 | 78.40 | 91.01 | 85.60 |
| Gradient-based mask | 89.39 | 74.59 | 60.30 | 84.80 | 63.49 | 76.40 | 74.83 |

lection.

## 4.7. Training Time and GPU Memory

We measured training time and peak GPU memory across all five models evaluated in Section 4.2. Table 5 reports these measurements. FrameFT trains 5-7% faster than LoRA and within 0.6 GiB of its peak GPU memory. The modest memory gap reflects the fact that the frozen pretrained backbone dominates the total GPU footprint. Both these measurements use standard PyTorch without custom kernels. Optimized kernels will be available on the GitHub repository shortly.

*Table 5.* Training time (minutes) and peak GPU memory (GiB) for LoRA and FrameFT on the Alpaca instruction tuning task.

| Metric | Method | Llama-2-7B | Llama-2-13B | Gemma-2-9B | Gemma-2-2B | Llama-3.1-8B |
|---|---|---|---|---|---|---|
| Training time (min) | LoRA | 238 | 363 | 323 | 126 | 231 |
| | FrameFT | 225 | 345 | 301 | 115 | 212 |
| GPU memory (GiB) | LoRA | 30.1 | 34.8 | 37.8 | 26.9 | 38.2 |
| | FrameFT | 29.5 | 34.7 | 37.7 | 26.8 | 37.6 |

## 5. Related work

Model adaptation for downstream tasks has been studied extensively in recent years, and has provided various efficient methods that reduce computation and storage needs while maintaining performance. Here, we describe different variants of PEFT methods briefly introduced in §1.

**Adapters:** Adapters introduce specialized modules between pre-existing layers within the pretrained model. These adapter layers are trained during fine-tuning while keeping the pretrained model parameters frozen (Houlsby et al., 2019; Karimi Mahabadi et al., 2021; He et al., 2022). By keeping the parameters of the original model frozen, they preserve the knowledge acquired during pretraining while reducing the risk of overfitting.

**Low-Rank Matrix Factorization:** LoRA techniques reparameterize the weight updates on selected layers of the model through low-rank factorizations (Hu et al., 2022). This framework of training only the decomposition matrices while freezing pretrained parameters has led to many variants exploring asymmetric chaining (Malinovsky et al., 2024), quantization-aware formulations (Dettmers et al., 2023), different learning rates for the update terms (Hayou et al., 2024) , and compressing the adapters into a single-matrix update (Bensaïd et al., 2025).

**Prefix and Prompt Tuning.** Prefix-tuning strategies prepend learnable vector sequences before transformer layer inputs, creating controllable input modifications while keeping the pretrained model fixed (Wang et al., 2025; Li & Liang, 2021; Qin & Eisner, 2021; Lester et al., 2021b; Liu et al., 2022b). Closely related, Prompt-tuning approaches learn different prompt representations to guide model behaviors without architectural modification (Xiao et al., 2025; Lester et al., 2021a; Liu et al., 2022c; Ge et al., 2022). Unlike prefix tuning, prompt tuning operates exclusively at the input embedding level, making it particularly efficient and easy to integrate (Lester et al., 2021a). This approach has been widely extended to domain adaptation (Ge et al., 2022), vision-language models (Zhou et al., 2022), and diffusion models (Dong et al., 2023; Chung et al., 2023).

**Sparse Fine-Tuning:** Sparse fine-tuning methodologies exploit natural parameter redundancy by targeting only critical components while freezing the rest (Iurada et al., 2025; Khaki et al., 2025; He et al., 2025; Lu et al., 2024; Gao et al., 2024; Guo et al., 2021). These approaches, whether through low-rank operations (Lu et al., 2024; Dragomir et al., 2026) or spectral compression via Fourier transformations (Gao et al., 2024; Zhang et al., 2026) – achieve competitive/superior performance compared to full fine-tuning but reducing trainable parameter count.

## 6. Conclusions

We describe FrameFT, a parameter-efficient fine-tuning framework that leverages structured subspace decompositions based on Fusion Frames to fine-tune transformer models for vision and language tasks. Our extensive empirical validation across both vision transformers and state-of-the-art language models (including the Llama and Gemma families), shows that substantial compute and parameter efficiency gains are achievable without sacrificing performance across many evaluation benchmarks. We also provide analysis showing that FrameFT preserves the Lipschitz smoothness of the loss landscape, and so achieves desirable convergence properties. We believe that one key advantage of parameter efficiency of FrameFT will be in situations where the use case requires a *set* of fine-tuned models, each fine-tuned on a specific task and invoked on a case-by-case basis. We note that support for structured sparsity (beyond $2:4$ sparsity) remains limited but this provides a concrete opportunity for higher efficiency gains if specialized kernels are implemented. The code is available at `https://github.com/vsingh-group/FrameFT`.

## Impact Statement

This paper presents work whose goal is to advance the field of Machine Learning by introducing more efficient methods for adapting foundation models. FrameFT aims to lower the computational cost and storage requirements for fine-tuning, which can facilitate broader accessibility and reduced energy usage in AI applications. We are not aware of any specific negative societal consequences or ethical issues unique to this work that require further highlighting.

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

# Appendix

In this appendix, we provide proofs for the theorems and additional details on the experiments presented in the main paper. In Section A, we provide a detailed proof for the lemma showing that FrameFT preserves the Lipschitz smoothness of the loss function. Section B presents the results for the convergence guarantee of Gradient Descent to a stationary point for Lipschitz smooth functions. Section C evaluates the performance of FrameFT for Image classification tasks. In Section D, we measure the performance of FrameFT as we vary the subspace dimension. In Section E, we show the performance of FrameFT with an increasing number of non-zero coefficients. We observe that FrameFT can leverage the additional parameters to improve the performance. We provide some additional discussions in Section F on the Frames and different subspace dimensions in Fusion Frames. Section G lists the hyperparameters used for our experiments. In Section H, we describe an algorithm for constructing Tight Fusion Frames along with an example. Finally, in Section I, we analyze the sensitivity of FrameFT to the scaling factor $\alpha$ and observe that FrameFT performs well above the baselines across a wide range of $\alpha$ values.

## A. FrameFT preserves Lipschitz smoothness

Here, for working through the proof, assume the following dimensions for the matrices involved: $P_m$ is $m \times p$, $P_n$ is $n \times q$, $C_l^j$ is $p \times q$ and $W_0$ is $m \times n$.

**Proof.** Let us consider how the loss function behaves at two different points. Take any two coefficient matrices $C_l^1$ and $C_l^2$. From our composition rule, these map to: $W^1 = W_0 + P_m C_l^1 P_n^T$ and $W^2 = W_0 + P_m C_l^2 P_n^T$. Scalar is omitted for notation simplicity.

By the $L$-Lipschitz smoothness assumption on $f(W)$:

$$||\nabla f(W^1) - \nabla f(W^2)||_F \leq L||W^1 - W^2||_F$$

where we use the notation $\nabla f(W) = \frac{\partial f}{\partial W}$, the derivative of $f$ with respect to $W$. We also have $W^1 - W^2 = P_m(C_l^1 - C_l^2)P_n^T$ and we will show shortly that $||W^1 - W^2||_F$ is upper-bounded by terms involving $(C_l^1 - C_l^2)$ and other constants. The chain rule for matrix derivatives means the derivative of $f$ with respect to the coefficients $C_l$ is,

$$\nabla f(C_l) = \frac{\partial f}{\partial W} \cdot \frac{\partial W}{\partial C_l} = \nabla f(W) \cdot \nabla W(C_l).$$

The term $\nabla W(C_l)$ denotes how the weight matrix $W$ changes with respect to a change in the coefficient matrix $C_l$. We will use directional derivatives to compute this.

Consider a matrix $\zeta_\Delta \in \mathbb{R}^{k\rho_m \times k\rho_n}$ which represents a direction. The directional derivative of $W$ with respect to $C_l$ in the direction of $\zeta_\Delta$ is given by

$$\nabla_{\zeta_\Delta} W(C_l) = \lim_{t \to 0} \frac{W(C_l + t\zeta_\Delta) - W(C_l)}{t} = \lim_{t \to 0} \frac{P_m \zeta_\Delta P_n^T t}{t} = P_m \zeta_\Delta P_n^T$$

Hence, the derivative of $f$ with respect to $C_l$ in the direction of $\zeta_\Delta$ is given by

$$\begin{aligned}
\nabla_{\zeta_\Delta} f(C_l) &= \nabla f(W) \cdot \nabla_{\zeta_\Delta} W(C_l) \\
&= \text{Tr}((\nabla f(W))^T (P_m \zeta_\Delta P_n^T)) \\
&\overset{(a)}{=} \text{Tr}(P_n^T \nabla f(W)^T P_m \zeta_\Delta) \\
&\overset{(b)}{=} \text{Tr}((P_m^T \nabla f(W) P_n)^T \zeta_\Delta) \\
&= \langle P_m^T \nabla f(W) P_n, \zeta_\Delta \rangle
\end{aligned}$$

The equality (a) follows from $\text{Tr}(AB) = \text{Tr}(BA)$ and the equality (b) follows from $(ABC)^T = C^T B^T A^T$.

From the definition of the directional gradient, we have that $\nabla_{\zeta_\Delta} f(C_l) = \langle \nabla f(C_l), \zeta_\Delta \rangle$. But from above, we have $\nabla_{\zeta_\Delta} f(C_l) = \text{Tr}((P_m^T \nabla f(W) P_n)^T \zeta_\Delta)$. These two expressions are equal for all $\zeta_\Delta$. Therefore,

$$\nabla f(C_l) = P_m^T \nabla f(W) P_n$$

Hence, for two coefficient matrices $C_l^1, C_l^2$:

$$
\begin{aligned}
||\nabla f(C_l^1) - \nabla f(C_l^2)||_F &= ||P_m^T(\nabla f(W^1) - \nabla f(W^2))P_n||_F \\
&\leq ||P_m||_{op}||(\nabla f(W^1) - \nabla f(W^2))P_n||_F \\
&\leq ||\nabla f(W^1) - \nabla f(W^2)||_F ||P_m||_{op}||P_n||_{op} \\
&\leq L||W^1 - W^2||_F \sqrt{B_m}\sqrt{B_n}
\end{aligned}
\tag{8}
$$

We can bound the operator norm of $P_m$ using the upper frame bounds.

$$
||P_m^T x||_2^2 \leq B_m ||x||^2 \implies ||P_m||_{op} = ||P_m^T||_{op} \leq \sqrt{B_m}
$$

Now, we need to bound $||W^1 - W^2||_F$. We know that $W^1 = W_0 + P_m C_l^1 P_n^T$ and $W^2 = W_0 + P_m C_l^2 P_n^T$. Therefore:

$$
W^1 - W^2 = P_m(C_l^1 - C_l^2)P_n^T
\tag{9}
$$

Now

$$
\begin{aligned}
||W^1 - W^2||_F &= ||P_m(C_l^1 - C_l^2)P_n^T||_F \\
&\overset{(a)}{\leq} ||P_m||_{op} \cdot ||(C_l^1 - C_l^2)P_n^T||_F \\
&\overset{(b)}{\leq} ||P_m||_{op} \cdot ||C_l^1 - C_l^2||_F \cdot ||P_n^T||_{op} \\
&= \sqrt{(B_m B_n)} \cdot ||C_l^1 - C_l^2||_F
\end{aligned}
\tag{10}
$$

(a) and (b) follow from $||AB||_F \leq ||A||_{op}||B||_F$ and $||AB||_F \leq ||A||_F||B||_{op}$ respectively. Substituting this bound in (8), we get

$$
\begin{aligned}
||\nabla f(C_l^1) - \nabla f(C_l^2)||_F &\leq \sqrt{B_m B_n} L\left(\sqrt{B_m B_n}||C_l^1 - C_l^2||_F\right) \\
&= L B_m B_n ||C_l^1 - C_l^2||_F
\end{aligned}
$$

Hence, $f(C_l)$ is $\tilde{L}$-Lipschitz smooth with $\tilde{L} = L B_m B_n$.

## B. Convergence Guarantee based on Lipschitz smoothness

The gradient descent update at iteration $t$ is: $C_l^{t+1} = C_l^t - \eta \nabla f(C_l^t)$.

Let $\tilde{L}$ be the Lipschitz constants derived in Section A. By $\tilde{L}$-Lipschitz smoothness:

$$
f(C_l^{t+1}) \leq f(C_l^t) + \langle \nabla f(C_l^t), C_l^{t+1} - C_l^t \rangle + (\tilde{L}/2)||C_l^{t+1} - C_l^t||_F^2
$$

Substituting $C_l^{t+1} = C_l^t - \eta \nabla f(C_l^t)$ we get:

$$
f(C_l^{t+1}) \leq f(C_l^t) + \langle \nabla f(C_l^t), -\eta \nabla f(C_l^t) \rangle + (\tilde{L}/2)\eta^2 ||\nabla f(C_l^t)||_F^2
\tag{11}
$$

$$
= f(C_l^t) - \eta ||\nabla f(C_l^t)||_F^2 + (\tilde{L}/2)\eta^2 ||\nabla f(C_l^t)||_F^2
\tag{12}
$$

Progress per step is

$$
f(C_l^t) - f(C_l^{t+1}) \geq \eta ||\nabla f(C_l^t)||_F^2 (1 - (\tilde{L}\eta)/2)
$$

Summing from $t = 0$ to $T - 1$:

$$
\sum_{t=0}^{T-1}[f(C_l^t) - f(C_l^{t+1})] \geq \eta(1 - (\tilde{L}\eta)/2) \sum_{t=0}^{T-1} ||\nabla f(C_l^t)||_F^2
$$

The left side telescopes:

$$f(C_l^0) - f(C_l^T) \geq \eta(1 - (\tilde{L}\eta)/2) \sum_{t=0}^{T-1} ||\nabla f(C_l^t)||_F^2$$

Since $f(C_l^T) \geq f^*$, we have:

$$f(C_l^0) - f^* \geq \eta(1 - (\tilde{L}\eta)/2) \sum_{t=0}^{T-1} ||\nabla f(C_l^t)||_F^2$$

Dividing both sides by $T$ and rearranging:

$$(1/T) \sum_{t=0}^{T-1} ||\nabla f(C_l^t)||_F^2 \leq (f(C_l^0) - f^*)/(\eta(1 - (\tilde{L}\eta)/2)T)$$

For $\eta \leq 1/\tilde{L}$, we have $1 - (\tilde{L}\eta)/2 \geq 1/2$, and so

$$(1/T) \sum_{t=0}^{T-1} ||\nabla f(C_l^t)||_F^2 \leq 2(f(C_l^0) - f^*)/(\eta T).$$

So, this descent reaches an $\epsilon$-first order stationary point.

## C. Image Classification with Vision Transformers

**Evaluation Framework:** We investigate the effectiveness of FrameFT for fine-tuning Vision Transformers. To this end, we fine-tune the base and large variants of the Vision Transformer (ViT) architecture (Dosovitskiy et al., 2021). We chose the ImageNet-21K pre-trained ViT models available on the Hugging Face Hub for our base model and evaluated across a diverse set of image classification challenges. The test suite includes fine-grained classification tasks – Oxford Pets (Parkhi et al., 2012), Stanford Cars, FGVC Aircraft (Maji et al., 2013), CIFAR10, CIFAR100, texture recognition – DTD (Cimpoi et al., 2014), and remote sensing applications – EuroSAT (Helber et al., 2018), RESISC45 (Cheng et al., 2017).

**FrameFT configuration:** For this experiment, we utilized sparse block diagonal coefficient matrices with $n = 1000$ and $n = 5000$ non-zero elements, and a scaling factor $\alpha = 250$ across all the tasks. We share the positions of non-zero positions across all the layers. Similar to the natural language understanding experiment, we set $\rho = 2$ while adjusting the number of subspaces $k$ to satisfy $k\rho = n$ throughout the network. Again, following LoRA, we adapt only the Query and Value matrices in the self-attention blocks in all layers of the model.

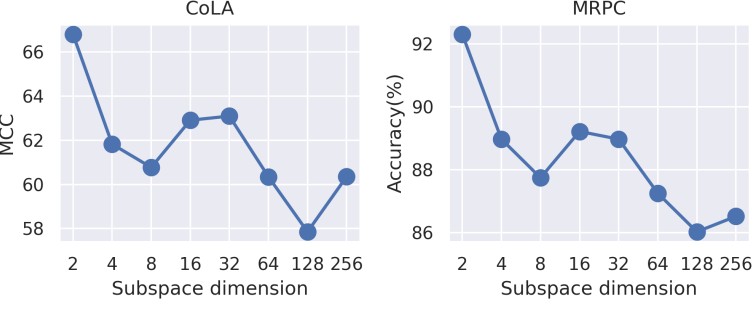

*Figure 5.* Performance of FrameFT on CoLA and MRPC for different values of subspace dimension $\rho$

**Performance analysis:** Table 6 shows the performance of FrameFT for fine-tuning Vision Transformers on diverse datasets. On average, with 5000 non-zero coefficients per layer, FrameFT performs better than all the baseline methods except full fine-tuning. FrameFT is able to achieve this with the lowest number of parameters compared to all the baseline methods, which is $5 - 20\times$ lower than LoRA. We also see an improvement in performance as we increase the number of non-zero coefficients from 1000 to 5000. As an additional note, the hyperparameters for FrameFT are held fixed for all the tasks in this benchmark. The hyperparameters for FourierFT were adjusted on a task-by-task basis, as noted by (Gao et al., 2024).

## D. Performance versus subspace dimension

In this experiment, we measure the performance of FrameFT for different values of subspace dimension, $\rho$. For this purpose, we fine-tune the RoBERTa-base model on two of the tasks in the GLUE benchmark - CoLA and MRPC, for different values of $\rho$. We keep the hyperparameters that we obtain for $\rho = 2$ as shown in Table 7. Figure 5 shows the performance of FrameFT as the subspace dimension is varied from 2 to 256. We observe that FrameFT performs well even with a subspace

*Table 6.* Performance of different methods for Fine-Tuning ViT Base and Large models on different datasets. * indicates the results reported in previous work. We highlight non PEFT methods in gray. FrameFT performs better than all the baseline methods on average, using a 30x lower number of parameters than LoRA.

| | Method | Params | Pets | Cars | CIFAR10 | DTD | EuroSAT | FGVC | RESISC45 | CIFAR100 | Avg. |
|---|---|---|---|---|---|---|---|---|---|---|---|
| ViT-B | Full finetune* | 85.8M | 93.14 | 79.78 | 98.92 | 77.68 | 99.05 | 54.84 | 96.13 | 92.38 | 86.49 |
| | LinearProbe* | – | 90.28 | 25.76 | 96.41 | 69.77 | 88.72 | 17.44 | 74.22 | 84.28 | 68.36 |
| | LoRA* | 581K | 93.19 | 45.38 | **98.78** | 74.95 | 98.44 | 25.16 | 92.70 | **92.02** | 77.58 |
| | FourierFT | 72K | 93.21 | 46.11 | 98.58 | 75.09 | 98.29 | 27.51 | 91.97 | 91.20 | 77.75 |
| | FourierFT | 239K | 93.05 | 56.36 | 98.69 | 77.30 | 98.78 | 32.44 | **94.26** | 91.45 | 80.29 |
| | FrameFT (ours) | 24K | **93.71** | 70.84 | 98.50 | 77.80 | 98.52 | 44.45 | 92.30 | 90.80 | 83.36 |
| | FrameFT (ours) | 120K | 93.55 | **78.16** | 98.50 | **79.96** | **98.91** | **52.35** | 94.10 | 90.90 | **85.80** |
| ViT-L | Full finetune* | 303.3M | 94.43 | 88.90 | 99.15 | 81.79 | 99.04 | 68.25 | 96.43 | 93.58 | 90.20 |
| | LinearProbe* | – | 91.11 | 37.91 | 97.78 | 73.33 | 92.64 | 24.62 | 82.02 | 84.28 | 72.96 |
| | LoRA* | 1.57M | **94.82** | 73.25 | **99.13** | 81.79 | 98.63 | 42.32 | 94.71 | **94.87** | 84.94 |
| | FourierFT | 144K | 94.46 | 69.56 | 99.10 | 80.83 | 98.65 | 39.92 | 93.86 | 93.31 | 83.71 |
| | FourierFT | 480K | 94.84 | 79.14 | 99.08 | **81.88** | 98.66 | 51.28 | **95.20** | 93.37 | 86.68 |
| | FrameFT (ours) | 48K | 94.27 | 78.48 | 99.02 | 79.94 | 98.49 | 52.63 | 93.72 | 92.68 | 86.15 |
| | FrameFT (ours) | 240K | 94.20 | **82.83** | 98.90 | 81.54 | **98.87** | **59.63** | 94.96 | 92.71 | **87.95** |

*Figure 6.* Performance of RoBERTa base model fine-tuned with FrameFT versus the number of non-zero coefficients on the GLUE benchmark. The red line indicates the performance of full fine-tuning.

dimension of 2. We do not see a strong relationship between the performance and the subspace dimension. Given that the two key choices we have for adjustments – the number of coefficients and the subspace dimension – to fill the space, the number of coefficients is a more consistent way of improving performance as indicated in Figure 6. An alternative and promising direction for future work is to make the subspace dimension a trainable parameter, potentially enabling more adaptive and efficient optimization.

## E. Performance as a function of number of non-zero coefficients

Figure 6 shows the performance of RoBERTa base fine-tuned with FrameFT on two of the tasks in the GLUE benchmark, in addition to the tasks shown in the main text. As noted earlier, we observe that FrameFT improves as the number of non-zero coefficients are increased, performing on par or better than full-finetuning.

## F. Additional Discussions

We cover a few relevant points not discussed in detail so far.

**(a)** *Comparison with Prompt Tuning.* Prior literature has extensively analyzed the trade-offs between prompt tuning and Parameter-Efficient Fine-Tuning (PEFT) methods (Wistuba et al., 2024; Pu et al., 2023). Notably, (Pu et al., 2023) provides a quantitative comparison of prompt tuning against methods like LoRA and $(IA)^3$, reporting that PEFT strategies generally yield superior performance. Since FrameFT demonstrates performance comparable to or exceeding

| Optimizer | AdamW |
|---|---|
| LR scheduler | Linear schedule with warmup |
| Batch size | 32 |
| Head learning rate | 3E-3 |
| Adapter learning rate | 0.12 |
| Warmup ratio | 0.06 |
| Max. Seq. length | 512 |
| $\alpha$ | 20.0 |

*Table 7.* Hyperparameters for the GLUE benchmark

| Optimizer | AdamW |
|---|---|
| LR scheduler | Linear schedule with warmup |
| Batch size | 50 |
| Head learning rate | 3E-2 |
| Adapter learning rate | 0.33 |
| Warmup ratio | 0.06 |
| $\alpha$ | 250.0 |

*Table 8.* Hyperparameters for the Image classification task

| Optimizer | AdamW |
|---|---|
| LR scheduler | Linear schedule with warmup |
| Batch size | 128 |
| Learning rate | 0.1 |
| Warmup ratio | 0.03 |
| $\alpha$ | 200.0 |

*Table 9.* Hyperparameters for the Instruction tuning task

that of LoRA, we infer that it maintains this advantage over prompt tuning. Hence, FrameFT serves as a more robust alternative to prompt tuning, delivering the reliability of PEFT methods while requiring a very low number of parameters.

**(b)** *Standard Frames versus Fusion Frames.* A natural question is whether standard Frames (characterized by a subspace dimension of 1) could be employed instead of Fusion Frames. While feasible in principle, the specific construction algorithm we outline in §3.2 enforces a minimum subspace dimension of 2. This constraint arises because we first generate the frame in the Complex domain; when transforming these components to the Real domain, the dimension inherently doubles. Consequently, if an application strictly requires a subspace dimension of 1 (while maintaining $k\rho = n$), the complexity of fusion frames is unnecessary. In such scalar cases, one can simply rely on classical orthonormal bases, such as Fourier bases or Wavelets.

**(c)** *How does performance vary as a function of subspace dimension?* We performed experiments changing the subspace dimension and report the results in §D. We observe that as the subspace dimension is increased (with $k\rho = n$), the sparsity in Fusion Frames increases, thereby reducing the degrees of freedom for the parameter update. This leads to a small drop in performance.

# G. Hyperparameters used for the experiments

We present the hyperparameters used for different experiments in Tables 7, 8, 9. As for the baselines, we use the hyperparameters suggested by the respective authors.

# H. Fusion Frames Construction: an example

In this section, we briefly outline an algorithm for generating Tight Fusion Frames with the help of an example. (Casazza et al., 2011) formalized a systematic framework for identifying the $(k, \rho, d)$ values for which a Tight Fusion Frame exists

and generating the TFF whenever it exists. Their algorithm generates a TFF in $\mathbb{C}^d$. Since we are mainly interested in real vector spaces, we use the simple extension in (Fickus et al., 2023b) to adapt the TFF to the real domain. The overall algorithm can be divided into three parts.

1. Construct a UNTF for $\mathbb{C}^\rho$ with $d$ elements by playing Spectral Tetris.
2. Modulate these vectors with $k^{\text{th}}$ roots of unity to form $k$ subspaces of dimension $\rho$ in $\mathbb{C}^d$.
3. Use the method in (Fickus et al., 2023b) to extend the result to real-valued spaces.

We describe these steps in detail by walking through the construction of a (6,3,11) TFF, a Tight Fusion Frame spanning $\mathbb{C}^{11}$ with $k = 6$ subspaces where each subspace has a dimension of $\rho = 3$.

## H.1. Spectral Tetris

The first step is to generate a "smaller" frame and in the next step, we modulate the smaller frame to generate a "larger" Tight Fusion Frame. After generating a TFF for $\mathbb{C}^d$ we can easily extend it to the Real Field by applying the entrywise map $x + iy \mapsto \begin{bmatrix} x & -y \\ y & x \end{bmatrix}$. So, $k = 6, \rho = 3, d = 11$. As the name suggests UNTFs are Tight frames where each frame vector has a unit norm. We construct a $4 \times 11$ matrix $F$ whose columns are the frame vectors for $\mathbb{C}^4$ which satisfies

In this first step, we generate a "smaller" frame - a Unit Norm Tight Frame (UNTF) for $\mathbb{C}^3$ with 11 vectors. We arrange these vectors in the columns of a matrix $F$. This UNTF is characterized by

- Columns with unit norm
- Rows are Orthogonal and have a constant norm, that is $FF^*$ is a constant multiple of the Identity matrix ( here the constant being $\frac{11}{3}$)

We start by filling the first two entries in $F$ with 1

$$F = \begin{bmatrix} 1 & 1 & ? & ? & ? & ? & ? & ? & ? & ? & ? \\ ? & ? & ? & ? & ? & ? & ? & ? & ? & ? & ? \\ ? & ? & ? & ? & ? & ? & ? & ? & ? & ? & ? \end{bmatrix}$$

The remaining norm left to be filled is $\frac{11}{3} - 2 = \frac{5}{3}$. We continue to fill in 1s until the required norm is less than 1. Here, we can do this only once, yielding

$$F = \begin{bmatrix} 1 & 1 & 1 & ? & ? & ? & ? & ? & ? & ? & ? \\ ? & ? & ? & ? & ? & ? & ? & ? & ? & ? & ? \\ ? & ? & ? & ? & ? & ? & ? & ? & ? & ? & ? \end{bmatrix}$$

This leaves a norm of $\frac{2}{3}$ to be filled. This can be added with a $2 \times 2$ matrix $T(x)$. $T(x)$ here is defined as follows:

$$T(x) := \frac{1}{\sqrt{2}} \begin{bmatrix} \sqrt{x} & \sqrt{x} \\ \sqrt{2-x} & -\sqrt{2-x} \end{bmatrix}, \qquad T(x)T^*(x) = \begin{bmatrix} x & 0 \\ 0 & 2-x \end{bmatrix}$$

After substituting $T(x)$ with $x = \frac{2}{3}$, $F$ is now

$$F = \begin{bmatrix} 1 & 1 & 1 & \frac{1}{\sqrt{3}} & \frac{1}{\sqrt{3}} & 0 & 0 & 0 & 0 & 0 & 0 \\ 0 & 0 & 0 & \frac{\sqrt{2}}{\sqrt{3}} & -\frac{\sqrt{2}}{\sqrt{3}} & ? & ? & ? & ? & ? & ? \\ 0 & 0 & 0 & 0 & ? & ? & ? & ? & ? & ? & ? \end{bmatrix}$$

Now, we continue adding ones in row two until the norm becomes less than 1 again.

$$F = \begin{bmatrix} 1 & 1 & 1 & \frac{1}{\sqrt{3}} & \frac{1}{\sqrt{3}} & 0 & 0 & 0 & 0 & 0 & 0 \\ 0 & 0 & 0 & \frac{\sqrt{2}}{\sqrt{3}} & -\frac{\sqrt{2}}{\sqrt{3}} & 1 & 1 & ? & ? & ? & ? \\ 0 & 0 & 0 & 0 & ? & ? & ? & ? & ? & ? & ? \end{bmatrix}$$

$$\begin{bmatrix}
1 & 1 & 1 & \frac{1}{\sqrt{3}} & \frac{1}{\sqrt{3}} & 0 & 0 & 0 & 0 & 0 & 0 \\
1 & \omega & \omega^2 & \frac{\sqrt{1}}{\sqrt{3}}\omega^3 & \frac{\sqrt{1}}{\sqrt{3}}\omega^4 & 0 & 0 & 0 & 0 & 0 & 0 \\
1 & \omega^2 & \omega^4 & \frac{\sqrt{1}}{\sqrt{3}} & \frac{\sqrt{1}}{\sqrt{3}}\omega^2 & 0 & 0 & 0 & 0 & 0 & 0 \\
1 & \omega^3 & 1 & \frac{\sqrt{1}}{\sqrt{3}}\omega^3 & \frac{\sqrt{1}}{\sqrt{3}} & 0 & 0 & 0 & 0 & 0 & 0 \\
1 & \omega^4 & \omega^2 & \frac{\sqrt{1}}{\sqrt{3}} & \frac{\sqrt{1}}{\sqrt{3}}\omega^4 & 0 & 0 & 0 & 0 & 0 & 0 \\
1 & \omega^5 & \omega^4 & \frac{\sqrt{1}}{\sqrt{3}}\omega^3 & \frac{\sqrt{1}}{\sqrt{3}}\omega^2 & 0 & 0 & 0 & 0 & 0 & 0 \\
0 & 0 & 0 & \frac{\sqrt{2}}{\sqrt{3}} & -\frac{\sqrt{2}}{\sqrt{3}} & 1 & 1 & \frac{1}{\sqrt{6}} & \frac{1}{\sqrt{6}} & 0 & 0 \\
0 & 0 & 0 & \frac{\sqrt{2}}{\sqrt{3}}\omega^3 & -\frac{\sqrt{2}}{\sqrt{3}}\omega^4 & \omega^5 & 1 & \frac{1}{\sqrt{6}}\omega & \frac{1}{\sqrt{6}}\omega^2 & 0 & 0 \\
0 & 0 & 0 & \frac{\sqrt{2}}{\sqrt{3}} & -\frac{\sqrt{2}}{\sqrt{3}}\omega^2 & \omega^4 & 1 & \frac{1}{\sqrt{6}}\omega^2 & \frac{1}{\sqrt{6}}\omega^4 & 0 & 0 \\
0 & 0 & 0 & \frac{\sqrt{2}}{\sqrt{3}}\omega^3 & -\frac{\sqrt{2}}{\sqrt{3}} & \omega^3 & 1 & \frac{1}{\sqrt{6}}\omega^3 & \frac{1}{\sqrt{6}} & 0 & 0 \\
0 & 0 & 0 & \frac{\sqrt{2}}{\sqrt{3}} & -\frac{\sqrt{2}}{\sqrt{3}}\omega^4 & \omega^2 & 1 & \frac{1}{\sqrt{6}}\omega^4 & \frac{1}{\sqrt{6}}\omega^2 & 0 & 0 \\
0 & 0 & 0 & \frac{\sqrt{2}}{\sqrt{3}}\omega^3 & -\frac{\sqrt{2}}{\sqrt{3}}\omega^2 & \omega^1 & 1 & \frac{1}{\sqrt{6}}\omega^5 & \frac{1}{\sqrt{6}}\omega^4 & 0 & 0 \\
0 & 0 & 0 & 0 & 0 & 0 & 0 & \frac{5}{\sqrt{6}} & -\frac{5}{\sqrt{6}} & 1 & 1 \\
0 & 0 & 0 & 0 & 0 & 0 & 0 & \frac{5}{\sqrt{6}}\omega & -\frac{5}{\sqrt{6}}\omega^2 & \omega^3 & \omega^4 \\
0 & 0 & 0 & 0 & 0 & 0 & 0 & \frac{5}{\sqrt{6}}\omega^2 & -\frac{5}{\sqrt{6}}\omega^4 & 1 & \omega^2 \\
0 & 0 & 0 & 0 & 0 & 0 & 0 & \frac{5}{\sqrt{6}}\omega^3 & -\frac{5}{\sqrt{6}} & \omega^3 & 1 \\
0 & 0 & 0 & 0 & 0 & 0 & 0 & \frac{5}{\sqrt{6}}\omega^4 & -\frac{5}{\sqrt{6}}\omega^2 & 1 & \omega^4 \\
0 & 0 & 0 & 0 & 0 & 0 & 0 & \frac{5}{\sqrt{6}}\omega^5 & -\frac{5}{\sqrt{6}}\omega^4 & \omega^3 & \omega^2
\end{bmatrix}$$

*Table 10.* $(\mathbf{6,3,11})$**-TFF** for $\mathbb{C}^{11}$. Here, $\omega = e^{i\pi/3}$. A pair of rows belongs to the same subspace if their indices differ by a multiple of 6

Now we insert $T(x)$ with the remaining norm. We repeat this process until all the rows are filled. The Final $F$ is given by

$$F = \begin{bmatrix}
1 & 1 & 1 & \frac{1}{\sqrt{3}} & \frac{1}{\sqrt{3}} & 0 & 0 & 0 & 0 & 0 & 0 \\
0 & 0 & 0 & \frac{\sqrt{2}}{\sqrt{3}} & -\frac{\sqrt{2}}{\sqrt{3}} & 1 & 1 & \frac{1}{\sqrt{6}} & \frac{1}{\sqrt{6}} & 0 & 0 \\
0 & 0 & 0 & 0 & 0 & 0 & 0 & \frac{5}{\sqrt{6}} & -\frac{5}{\sqrt{6}} & 1 & 1
\end{bmatrix}$$

## H.2. Modulation

In the second step of TFF construction, the $F$ matrix is modulated with complex roots of unity, one subspace at a time. For each $k_i = 0, 1, 2, \ldots k-1$, we construct a row vector

$$w_{k_i} = \left[ \left( e^{\frac{i2\pi k_i}{k}} \right)^0 \left( e^{\frac{i2\pi k_i}{k}} \right)^1 \left( e^{\frac{i2\pi k_i}{k}} \right)^2 \ldots \left( e^{\frac{i2\pi k_i}{k}} \right)^{d-1} \right]$$

Each row of $F$ is multiplied by $w_{k_i}$ to produce the orthogonal basis for the subspace indexed by $k_i$. Theorem 14 by Casazza et al. (2011) proves that the Fusion Frames generated by this algorithm are Tight. The Final Fusion Frame generated is shown in Table 10.

## H.3. Fusion Frames construction time

Table 11, presents the time taken to construct the Fusion Frames using the Spectral Tetris method. As a reference, we provide the time taken to construct a random orthonormal basis using two methods, namely the Gram-Schmidt algorithm and QR decomposition on a random matrix. These times are measured on an NVIDIA A100 machine. Since the Gram-Schmidt process is sequential, it cannot be parallelized on GPUs/TPUs. Hence, the time taken to perform this process is quite high. We can see that Spectral Tetris takes a fraction of the time to construct than QR decomposition on modern GPUs. Though there is some latency in generating Fusion Frames, it is a minor issue, as it is amortized across multiple model calls.

*Table 11.* Basis generation time. Spectral Tetris takes a fraction of time to construct the Basis on an NVDIA A100 gpu

| Model | Method | Bases construction time (s) |
|---|---|---|
| | Gram-Schmidt | 112.1 |
| Llama-2-7B | QR decomposition | 0.31 |
| | Spectral Tetris | 0.09 |
| | Gram-Schmidt | 228.8 |
| Gemma-2-9B | QR decomposition | 0.64 |
| | Spectral Tetris | 0.21 |
| | Gram-Schmidt | 119.6 |
| Llama-3.1-8B | QR decomposition | 0.53 |
| | Spectral Tetris | 0.11 |

*Table 12.* Scaling factor $\alpha$ sensitivity on Llama-2-7B/Alpaca. FrameFT outperforms LoRA (63.18) across the full sweep.

| $\alpha$ | ARC-c | ARC-e | BoolQ | HellaSwag | OBQA | PIQA | RTE | WinoGrande | Avg. |
|---|---|---|---|---|---|---|---|---|---|
| 10 | 45.9 | 77.4 | 78.75 | 58.01 | 34.0 | 78.67 | 61.01 | 70.0 | 62.97 |
| 100 | 45.47 | 77.1 | 78.53 | 58.28 | 34.4 | 78.78 | 63.89 | 70.71 | 63.40 |
| 200 (default) | 45.22 | 76.93 | 78.62 | 58.08 | 34.2 | 78.62 | 66.06 | 71.19 | 63.62 |
| 400 | 45.47 | 77.31 | 79.2 | 58.32 | 34.0 | 78.62 | 62.45 | 70.32 | 63.21 |
| 600 | 45.56 | 77.02 | 78.99 | 58.31 | 34.6 | 78.73 | 62.82 | 70.56 | 63.32 |
| LoRA | 45.82 | 77.02 | 78.81 | 58.08 | 35.2 | 78.83 | 61.01 | 70.72 | 63.18 |
| Full FT | 47.52 | 77.73 | 78.96 | 58.99 | 33.6 | 78.61 | 62.09 | 69.61 | 63.39 |

## I. Scaling Factor Sensitivity

Table 12 reports performance of Llama-2-7B instruction-tuned on the Alpaca dataset with FrameFT as the scaling factor $\alpha$ is varied from 10 to 600. FrameFT outperforms LoRA's average of 63.18 across the entire sweep. For $\alpha \geq 100$, it consistently matches or exceeds the full fine-tuning upper bound of 63.39. The results demonstrate a major practical advantage: FrameFT is highly insensitive to this factor. Unlike PEFT methods requiring exhaustive searches to prevent instability, FrameFT operates reliably out-of-the-box.

