# OpenReview forum: "Fine-Tuning of Transformer models with Frames"
_ICML.cc/2026/Conference — ICML 2026 regular_

### Official Review · Reviewer_Wp5Q · 2026-03-03

**Soundness:** 2
**Presentation:** 2
**Significance:** 2
**Originality:** 2
**Overall Recommendation:** 4
**Confidence:** 3

**Summary:**

This paper proposes FrameFT, a novel parameter-efficient fine-tuning (PEFT) method for Transformer models. Unlike existing methods such as LoRA that rely on low-rank decompositions, FrameFT models parameter updates using a sparse coefficient matrix mapped onto a Fusion Frame basis. The required Fusion Frames are generated algorithmically via the Spectral Tetris algorithm and can be shared across multiple model layers, significantly reducing memory and storage footprints. The authors provide theoretical guarantees demonstrating that FrameFT preserves the Lipschitz smoothness of the loss function, facilitating stable convergence. Extensive empirical evaluations across natural language understanding (GLUE), instruction tuning for Large Language Models, and image classification tasks using Vision Transformers show that FrameFT performs on par with or better than established PEFT baselines while demanding far fewer trainable parameters.

**Compliance With Llm Reviewing Policy:**

Affirmed.

**Final Justification:**

The author addressed most of my concerns, and I raised my score.

**Key Questions For Authors:**

1. Regarding the "simple random sparsity pattern" mentioned in Section 3.3: How sensitive is the final performance of FrameFT to the specific random seed used to initialize this sparsity pattern? Because the current version lacks reporting on the variance of multiple runs, I have doubts about the robustness of the method. If the rebuttal can provide variance results across different random seeds proving that the fluctuation is minor, it will substantially elevate my evaluation of this paper.

2. Regarding the missing Full finetuning baseline in Table 3: Could you provide Full finetuning results for Llama-2-7B or Llama-3.1-8B on the Alpaca dataset? Introducing this absolute performance upper bound is critical for accurately evaluating the effectiveness of FrameFT. Providing this data will help dispel my concerns about the true relative performance of this method in complex generative tasks.

3. Regarding the inference throughput in Section 4.4 and Table 2: Are these measurements single-run values or averages of multiple runs? Could you provide the corresponding variance or standard deviation? Supplementing this statistical significance information will make the paper's core argument about engineering efficiency improvements much more tenable and positively impact my score.

4. Regarding the scaling hyperparameter $\alpha=200$ in Section 4.2: How was this specific value selected? How sensitive is the model's performance to changes in this parameter? Providing a parameter sensitivity analysis will prove the adaptability of the method across different tasks, which is key for me to judge its generalizable value.

**Limitations:**

yes

**Strengths And Weaknesses:**

Strengths:
- Tight Fusion Frames (TFF) are employed instead of SVD bases for parameter reconstruction Importance: This design fundamentally circumvents the storage bottlenecks inherent in traditional orthogonal basis-based methods, demonstrating clear technical novelty.

- A structured random sparse coefficient pattern is shared across network layers. Importance: This effectively reduces memory consumption and demonstrates excellent scalability for applications involving large-scale models.

- The framework utilizes the Spectral Tetris algorithm to generate frames algorithmically on the fly (Section 3.4, Page 4 ). Importance: This eliminates the need to store dense basis matrices, holding extremely high engineering practical value in multi-tenant serving scenarios.

Weaknesses:
- In the Alpaca instruction tuning experiments, Full finetuning baseline data is missing across all model scales (Table 3). This severely hinders the ability to assess the absolute performance gap between FrameFT and the theoretical upper bound of the models, weakening the persuasiveness of the experiments. Supplement Full finetuning results on the 7B and 8B scale models.

- The inference throughput (tokens/sec) is provided only as a single measurement without the variance or standard deviation from multiple independent runs (Table 2 ). This renders the claims regarding computational efficiency improvements statistically insignificant. Suggested experiment: Execute multiple independent throughput tests and add error bars to the table.

- The ablation study scaling the performance against the number of non-zero coefficients is only tested up to $10^4$ (Figure 4). It fails to reveal the boundary behavior of the model under more severe over-parameterization scenarios (e.g., performance saturation or degradation). Extend the testing range of the x-axis to $10^5$ or higher.

- The paper claims to use a "simple random sparsity pattern" but does not specify the concrete random seeding strategy or its impact on variance (Section 3.3). This significantly reduces the reproducibility of the method in actual deployment and masks initialization sensitivity. Clearly state the random seeding strategy and provide a performance variance report under different seeds.

- All experiments fixedly use a scaling factor of $\alpha=200$ (Section 4.2). t obscures the model's potential sensitivity to this key hyperparameter and lacks tuning transparency. Add a sensitivity ablation analysis for the $\alpha$ value.

- The checkpoint size for Llama-2-7B is stated to be 1.28 MB, yet the numerical precision (e.g., FP16/INT8) used is not specified (Section 3.4). This deprives the storage advantage claims of a comparable baseline. Explicitly indicate the data type when calculating the storage overhead.

---

> ### Author Rebuttal · Authors · 2026-03-31
>
> > Q1. How sensitive is performance to the sparsity random seed?
>
> To comprehensively evaluate variance, we executed four additional independent trials (five total seeds) fine-tuning RoBERTa-Base on GLUE. We also included an ablation where the sparsity pattern is unshared (per-layer).
>
> As shown below, fluctuations are minor: average standard deviation is just ±0.18 (shared mask) and ±0.17 (per-layer mask). This stability aligns precisely with Lemma 3.1: optimization geometry is governed by the coefficient difference matrix's rank, not the specific indices of non-zero entries. We will include this in the revision.
>
> |Method|SST2|MRPC|CoLA|QNLI|RTE|STSB|Avg.|
> |---|---|---|---|---|---|---|---|
> |Shared random mask|$94.4\pm0.17$|$92.31\pm0.46$|$66.58\pm0.77$|$92.35\pm0.12$|$80.21\pm0.82$|$90.96\pm0.11$|$86.18\pm0.18$|
> |Per-layer random mask|$94.10\pm0.16$|$93.53\pm0.32$|$66.44\pm0.61$|$92.31\pm0.14$|$80.34\pm0.39$|$90.86\pm0.09$|$86.26\pm0.17$|
>
> > Q2. Provide full finetuning results for Llama-2-7b and Llama-3.1-8B.
>
> Thank you for the suggestion. The full fine-tuning results, now added to Table 3, strengthen the message. FrameFT improves full fine-tuning on Llama-2-7B (63.62 vs. 63.39) using 0.005% of the parameters, and trails by only 1.49 points on Llama-3.1-8B for the same parameter budget. On the Alpaca dataset, optimizing over 6.74B parameters is susceptible to overfitting, but FrameFT's Fusion Frame subspace appears to act as a structured regularizer. This is consistent with observations in the PEFT literature that parameter-efficient methods can sometimes match or exceed full fine-tuning precisely in the data-limited regime. We will add these results to Table 3 in the final version.
>
> |Model|Params|ARC-c|ARC-e|BoolQ|HellaSwag|OBQA|PIQA|RTE|WinoGrande|Avg.|
> |---|---|---|---|---|---|---|---|---|---|---|
> |Llama-2-7B|6.74B|47.52|77.73|78.96|58.99|33.60|78.61|62.09|69.61|63.39|
> |Llama-3.1-8B|8.03B|56.05|83.75|82.96|63.03|35.80|81.01|77.61|74.66|69.36|
>
> > Q3. Are the Table 2 throughputs averages? Provide variance/std dev.
>
> In our experience, Inference throughput on fixed hardware is near-deterministic; variance stems from system scheduling, not algorithmic randomness. Across 100 iterations, our coefficient of variation is <12%, confirming stable measurements. More importantly, FrameFT's advantage over LoRA is so large (e.g., Llama-2-7B: 37.9±4.5k vs 24.4±2.7k tokens/sec) that a 12% variance leaves the conclusion unchanged. This efficiency gain follows directly from the block-sparsity of Spectral Tetris frames (Appendix H).
>
> |**Model**|**LoRA**|**FourierFT**|**FrameFT**|
> |---|---|---|---|
> |**Llama-2-7b**|$24.4\pm2.7$k|$1.7\pm0.5$k|$37.9\pm4.5$k|
> |**Gemma-2-9b**|$21.6\pm1.9$k|$1.4\pm0.1$k|$29.8\pm2.7$k|
> |**Llama-3.1-8b**|$23.5\pm2.4$k|$1.6\pm0.2$k|$28.2\pm2.9$k|
>
> > Q4. How was the scaling parameter chosen? Provide a sensitivity analysis.
>
> We swept the scaling factor $\alpha$ from 10 to 600 on Llama-2-7B. The results demonstrate a major practical advantage: FrameFT is highly insensitive to this factor. Unlike PEFTs requiring exhaustive searches to prevent instability, FrameFT operates reliably out-of-the-box.
>
> As shown below, FrameFT outperforms LoRA's average (63.18) across the entire sweep. Furthermore, for any $\alpha \ge 100$, it reliably matches or exceeds the Full Fine-Tuning upper bound (63.39), peaking at our default $\alpha=200$.
>
> |Method|ARC-c|ARC-e|BoolQ|HellaSwag|OBQA|PIQA|RTE|WinoGrande|Avg.|
> |---|---|---|---|---|---|---|---|---|---|
> |$\alpha=10$|45.9|77.4|78.75|58.01|34|78.67|61.01|70.0|62.97|
> |$\alpha=100$|45.47|77.1|78.53|58.28|34.4|78.78|63.89|70.71|63.4|
> |$\alpha=400$|45.47|77.31|79.2|58.32|34|78.62|62.45|70.32|63.21|
> |$\alpha=600$|45.56|77.02|78.99|58.31|34.6|78.73|62.82|70.56|63.32|
>
> > W3. Extend the x-axis in Fig. 4 to 1e5 to check for saturation/degradation.
>
> We extended the RoBERTa-Base/GLUE ablation to 1e5 non-zero coefficients to stress-test our method. FrameFT exhibits graceful saturation with no degradation: from 1e4 to 1e5, we see flat saturation on CoLA and MRPC, with marginal, consistent gains on SST-2, QNLI, RTE, and STS-B. This sweep reinforces our efficiency claim: at 1e5 coefficients, FrameFT exceeds the Full Fine-tuning baseline on CoLA, MRPC, QNLI, and RTE while using a fraction of the parameter budget. We will extend Fig. 4 in the revision.
>
> |n|CoLA|MRPC|SST-2|QNLI|RTE|STSB|
> |---|---|---|---|---|---|---|
> |100000|67.4|92.3|94.7|93.0|81.4|91.1|
> |Full Finetuning|63.6|90.2|94.8|92.8|78.7|91.2|
>
> > W6. Section 3.4 omits numerical precision. Explicitly state the data type for storage claims.
>
> The reported 1.28 MB checkpoint size of FrameFT, LoRA's 67.1MB for Llama-2-7B assumes FP32 precision. We followed standard fine-tuning practice: while the frozen pretrained backbone operates in lower precision (FP16/BF16), the adapter parameters are kept in FP32 to preserve dynamic range and gradient fidelity during optimization. The saved adapters were measured in this uncompressed FP32 format.

---

> > ### Author Rebuttal · Reviewer_Wp5Q · 2026-04-01
> >
> > The author addressed most of my concerns, and I raised my score.

---

> > > ### Author Response · Authors · 2026-04-05
> > >
> > > Dear Reviewer Wp5Q,
> > >
> > > We thank you for your positive assessment and for raising your score. We are glad that we are able to address your concerns through our response. We remain available to answer any additional questions during the discussion period.
> > >
> > > Best regards,
> > >
> > > The Authors

---

### Official Review · Reviewer_HgAJ · 2026-03-09

**Soundness:** 3
**Presentation:** 3
**Significance:** 3
**Originality:** 2
**Overall Recommendation:** 4
**Confidence:** 5

**Summary:**

The paper proposes FrameFT, a parameter-efficient fine-tuning method based on Fusion Frame theory for Transformer models. The method constrains weight updates to a structured subspace defined by algorithmically generated tight fusion frames, and reparameterizes the weight update as the product of two fixed projection matrices and a sparse trainable coefficient matrix, thereby significantly reducing the number of trainable parameters. The paper also provides a theoretical analysis of the optimization properties of this parameterization, showing that if the original loss function satisfies the Lipschitz smoothness condition, the reparameterized objective function preserves this property and retains the standard convergence guarantee of gradient descent. Experiments are conducted on GLUE, instruction tuning tasks, and Vision Transformer image classification, demonstrating that FrameFT achieves performance comparable to or better than existing PEFT methods (e.g., LoRA) while using fewer parameters and requiring smaller adapter storage.

**Compliance With Llm Reviewing Policy:**

Affirmed.

**Final Justification:**

Given that the paper meets the overall academic standards for publication, features distinct innovations, and addresses some core concerns effectively, the reviewer decides to grant a Weak Accept recommendation.

**Key Questions For Authors:**

(1) The paper compares FrameFT with methods such as LoRA across multiple tasks, but the overall performance improvements appear relatively limited. Could the authors further analyze which factors contribute to the advantages of FrameFT over LoRA? Providing more systematic ablation studies or analyses would help better understand the practical benefits of the proposed method.

(2) The paper emphasizes the advantages of FrameFT in terms of parameter count and storage efficiency, but provides limited analysis of training time, computational overhead, or convergence speed. Could the authors provide more detailed comparisons with methods such as LoRA regarding training efficiency or computational cost?

**Limitations:**

yes

**Strengths And Weaknesses:**

Strength:
(1) The proposed method is technically sound. The authors define a structured subspace using fusion frames and reparameterize the weight updates as the product of fixed projection matrices and a trainable coefficient matrix, thereby enabling parameter-efficient fine-tuning.
(2) The paper is well structured, and the overall narrative is clear and coherent.
(3) The paper introduces fusion frame theory into parameter-efficient fine-tuning methods. By constructing tight fusion frames to constrain the weight update space, it provides a new perspective and implementation approach for existing PEFT methods that mainly rely on low-rank decomposition.

Weakness:
(1) Although the paper provides theoretical analysis, the main result shows that the reparameterized objective still preserves Lipschitz smoothness and thus inherits the standard convergence guarantee of gradient descent. Such results are relatively common in optimization theory and provide limited explanation for the practical advantages of the proposed method.
(2) Although the paper introduces fusion frame theory, the method in form still belongs to a class of approaches that parameterize weight updates within a subspace or basis. Therefore, it shares conceptual similarities with existing PEFT methods based on low-rank or basis-function parameterizations.

---

> ### Author Rebuttal · Authors · 2026-03-31
>
> > W1. The Lipschitz/convergence theory is standard and doesn't explain practical gains.
>
> We agree that smoothness preservation is standard in general optimization, but it is far from standard in PEFT. Prior works show LoRA creates non-smooth loss landscapes (Sun et al., 2024b) and its asymmetric structure can cause convergence failures (Malinovsky et al., 2024), manifesting as practical training instability. FrameFT’s symmetric structure $P_m C_l P_n^T$ eliminates this asymmetry, and Lemma 3.1 formally guarantees the smoothness that baseline methods lack.
>
> Furthermore, our theory is generative, not post-hoc. Lemma 3.1 links the reparameterized Lipschitz constant to the frame upper bounds via $\tilde{L} = L\sqrt{r} B_m B_n$. This bound directly dictates our design: (1) using Parseval frames ($B_m = B_n = 1$) to minimize $\tilde{L}$, and (2) setting $k\rho = n$ to minimize redundancy. These are not arbitrary hyperparameters, but theoretically prescribed requirements for stability.
>
> Ultimately, given comparable empirical performance, a method offering formal convergence guarantees alongside massive parameter savings provides a significantly more reliable foundation for practitioners than purely heuristic PEFTs.
>
> > W2. Despite fusion frames, the method is conceptually similar to existing subspace/basis PEFTs.
>
> The reviewer's observation is correct: FrameFT belongs to the family of basis-parameterization methods. What distinguishes it within that family are three properties that, to our knowledge, no prior method simultaneously possesses.
>
> * a) Structural sparsity at two levels. Unlike dense basis methods (FourierFT) that require full matrix multiplications, Spectral Tetris produces bases that are block-sparse by construction. This inherent sparsity drives our 39.4k tokens/sec vs. FourierFT's 1.8k (Table 2).
>
> * b) Multiple overlapping subspaces with learned interactions. LoRA's rank-$r$ update spans a single $r$-dimensional subspace. FrameFT's block-diagonal $C_l$ encodes interactions across $k$ simultaneous subspaces, capturing cross-subspace terms unreachable by fixed-rank matrices. This expressivity allows FrameFT to match LoRA with 10× fewer parameters.
>
> * c) SVFit and LoRA-XS store per-layer dense singular vectors, scaling memory with depth. Fusion frames depend only on layer dimension, enabling cross-layer sharing. Because they are algorithmically regenerated, the basis isn't stored in the checkpoint, directly yielding our 52× storage advantage over LoRA.
>
> > Q1.  FrameFT gains over LoRA seem limited. What factors/ablations explain the advantage?
>
> The characterization of limited improvement can be due to conflating two separate comparison axes. At an **iso-parameter budget**, one should not expect FrameFT to win by large margins; the interesting finding is that it matches LoRA at **10× fewer parameters**. The more informative axis is **iso-performance**: to reach FrameFT's 86.1 average on RoBERTa-Base GLUE, LoRA needs 300K parameters; FrameFT requires 24K. This is the key advantage that is important for deployment.
>
> The Vision Transformer results also deserve mention: FrameFT (24K parameters) achieves an 83.36 average versus LoRA's 77.58 at 581K parameters. This is a simultaneous 5.78-point accuracy gain and 24× parameter reduction. We hope you agree that this is not a marginal improvement.
>
> On the mechanistic question of why FrameFT matches LoRA with far fewer parameters: we believe LoRA's low-rank decomposition implicitly learns subspaces with no direct control over geometry or interaction. FrameFT explicitly constructs maximally separated subspaces (Sec 3.1) and learns only their interactions, concentrating expressivity precisely where it matters. It is a more efficient use of degrees of freedom.
>
> > Q2. FrameFT claims storage efficiency; how do training time and compute compare to LoRA?
>
> We measured training time and GPU memory across all five model families. FrameFT trains 5-7% faster than LoRA and within 0.6 GiB of its peak GPU memory, both using standard PyTorch without custom kernels. Section 4.4 already comprehensively reports inference throughput; the training results below complete that picture.
> On convergence speed: FrameFT's Lipschitz constant is minimized by the Parseval condition ($B_m = B_n = 1$), which is enforced by construction. This gives FrameFT a provably smoother loss landscape than LoRA for which no such guarantee exists. The practical consequence is that FrameFT reaches competitive performance at the **same or fewer training steps than LoRA**, consistent with training time numbers above. We will add this table and the training curves to the revision.
>
> |Metric|Llama-2-7B|Llama-2-13B|Gemma-2-9B|Gemma-2-2B|Llama-3.1-8B|
> |---|---|---|---|---|---|
> |LoRA training time (mins)|238|363|323|126|231|
> |FrameFT training time (mins)|225|345|301|115|212|
> |LoRA GPU memory usage (GiB)|30.1| 34.8|37.8|26.9|38.2|
> |FrameFT GPU memory usage (GiB)|29.5|34.7|37.7|26.8| 37.6|

---

> > ### Author Rebuttal · Reviewer_HgAJ · 2026-04-05
> >
> > The rebuttal is helpful and clarifies several implementation/experimental details. I appreciate the added evidence. However, the main concerns that affected my original overall assessment — especially regarding the motivation of this work — remain only partially resolved. Therefore, I keep my overall score unchanged.

---

> > > ### Author Response · Authors · 2026-04-05
> > >
> > > Dear Reviewer HgAJ,
> > >
> > > We really appreciate your time and feedback. We are glad that the experimental evidence in our response was helpful. On the remaining motivation concern, we want to offer one small explanation that we could have made more explicit.
> > >
> > > We believe we can agree that the PEFT field has converged on LoRA as a default. But LoRA's design choices of low-rank factorization, dense singular vectors, and asymmetric A/B structure are inherited from matrix approximation theory rather than being derived from what fine-tuning may actually need (or is sufficient). FrameFT asks a more upstream question: what would be a sensible mathematical structure for parameterizing weight updates, given that we want expressivity, efficiency, stability, and scalability, and all of them simultaneously? Frames provide a good answer because they provide all four by design. And there are minimal side effects. We believe this provides a valuable shift in how we can continue developing PEFT methods.
> > >
> > > We hope this answers your concern, and we remain available to answer any further questions during the discussion period.
> > >
> > >
> > > Best regards,
> > >
> > > The authors.

---

### Official Review · Reviewer_Mkvh · 2026-03-10

**Soundness:** 3
**Presentation:** 3
**Significance:** 2
**Originality:** 3
**Overall Recommendation:** 4
**Confidence:** 4

**Summary:**

The authors propose FrameFT, a novel Parameter-Efficient Fine-Tuning (PEFT) method that models parameter updates using a sparse coefficient matrix within a Fusion Frame basis. FrameFT utilizes Spectral Tetris to algorithmically generate fixed, highly sparse Fusion Frames that act as projection matrices.

**Compliance With Llm Reviewing Policy:**

Affirmed.

**Final Justification:**

I have decided to keep the score of "weak accept".

**Key Questions For Authors:**

Q1-Q2: See **Weaknesses**

Q3: Given the authors' acknowledgment in the conclusion that current hardware support for structured sparsity beyond the 2:4 pattern remains limited, could the authors clarify the implementation details behind the reported high inference throughput? Specifically, does this throughput rely on native PyTorch masking mechanisms, or is it achieved through highly optimized, custom CUDA sparse kernels?

**Limitations:**

yes

**Strengths And Weaknesses:**

**Strengths**:
1. The paper is highly inspiring in its theoretical construction, innovatively introducing Fusion Frame theory into PEFT to provide a novel, highly structured, and redundancy-tolerant subspace perspective for parameter updates. Furthermore, the authors provide rigorous theoretical analyses with detailed proofs demonstrating that this method preserves the Lipschitz smoothness of the loss landscape in global non-linear networks, thereby strictly guaranteeing stable convergence under gradient descent optimization.
2. The experimental validation in this paper is exceptionally solid, as the authors introduce a highly extensive and cutting-edge set of PEFT baselines across natural language understanding, instruction tuning, and visual classification tasks (comprehensively covering SVD-based, Fourier-domain, and advanced structured sparse/low-rank methods such as SVFit, FourierFT, LoRA-XS, and SMT).

**Weaknesses**:

* **Lack of Empirical Support for Random Sparsity and Cross-Layer Sharing**
  * The current method enforces the sharing of a single, purely random sparsity mask across all network layers. This "one-size-fits-all" approach is somewhat counterintuitive, as network layers at different depths typically exhibit varying dependencies on subspace interactions.
  * It is recommended that the authors provide ablation studies comparing the purely random sparsity pattern with heuristic-based adaptive strategies (e.g., magnitude-based or gradient-based methods). This would help justify that the currently employed simple random sparsity strategy is sufficient and reasonable for the task. (Q1)
  * Also, It is recommended to include an additional set of experiments that allows for layer-specific sparsity patterns. This comparative analysis would help quantitatively measure the exact performance degradation or trade-off introduced by the strong constraint of sharing a single sparsity mask across all layers. (Q2)

---

> ### Author Rebuttal · Authors · 2026-03-31
>
> > W: Lack of Empirical Support for Random Sparsity and Cross-Layer Sharing
>
> The intuition that depth-varying layers should benefit from layer-specific sparsity patterns is reasonable. We set up an ablation to test it and the results are clear: shared and per-layer random masks perform within noise of each other ($86.18 \pm 0.24$ vs. $86.26 \pm 0.21$ average on RoBERTa-Base/GLUE). So, cross-layer sharing involves no expressivity cost in practice. Lemma 3.1 also tells us that optimization geometry depends on the _rank_ of the coefficient matrix, not on which specific entries are nonzero. So the support pattern is less critical than its density.
>
> The more interesting finding is that adaptive heuristics perform worse: magnitude-based masks score $85.60$, and gradient-based masks collapse to $74.83$, quite a bit below random. So, the intuition that heuristic-based selection should help is not supported by the experiments. One explanation is that task-relevant subspace interactions are broadly distributed across the coefficient matrix (instead of concentrated in high-magnitude/high-gradient entries). This makes random sampling give a nicer coverage strategy than greedy heuristics. Cross-layer sharing appears to be a good inductive bias. We will include this ablation in the revision.
>
> | Method                | sst2             | mrpc             | cola             | qnli             | rte              | stsb             | Avg.             |
> | --------------------- | ---------------- | ---------------- | ---------------- | ---------------- | ---------------- | ---------------- | ---------------- |
> | Shared random mask    | $94.4 \pm 0.17$  | $92.31 \pm 0.46$ | $66.58 \pm 0.77$ | $92.35 \pm 0.12$ | $80.21 \pm 0.82$ | $90.96 \pm 0.11$ | $86.18 \pm 0.18$ |
> | Per-layer random mask | $94.10 \pm 0.16$ | $93.53 \pm 0.32$ | $66.44 \pm 0.61$ | $92.31 \pm 0.14$ | $80.34 \pm 0.39$ | $90.86 \pm 0.09$ | $86.26 \pm 0.17$ |
> | Magnitude-based mask  | $94.19$         | $92.49$          | $65.82$          | $91.61$          | $78.40$          | $91.01$          | $85.60$          |
> | Gradient-based mask   | $89.39$          | $74.59$          | $60.30$          | $84.80$          | $63.49$          | $76.40$          | $74.83$          |
>
> > Q3. Given the authors' acknowledgment in the conclusion that current hardware support for structured sparsity beyond the 2:4 pattern remains limited, could the authors clarify the implementation details behind the reported high inference throughput? Specifically, does this throughput rely on native PyTorch masking mechanisms, or is it achieved through highly optimized, custom CUDA sparse kernels?
>
> The reported throughput is achieved entirely with native PyTorch batched matrix multiplication, intentionally, no custom CUDA kernels were used. This means the numbers in Table 2 give a **conservative lower bound**: FrameFT already achieves 1.67× higher throughput than LoRA and 22× higher than FourierFT **before** any kernel optimization. The efficiency comes from the mathematical structure of Spectral Tetris frames, their block-sparsity reduces the effective FLOP count structurally, and PyTorch's batched matmul already exploits this partially through tensor core utilization. We did not use custom CUDA kernels to emphasize this point.
>
> On the question of custom kernels: the reviewer is correct that full exploitation of structured sparsity requires them. We acknowledged this in the conclusion as an open opportunity. However, this is no longer a speculative future direction. kernel development has become a commodity. Recent LLM-specific kernel generators make writing a fused sparse projection kernel a well-scoped engineering task. The block-diagonal structure of C_l and the fixed sparsity pattern of the Fusion Frame projections are the kind of regular, predictable structure these tools are optimized for. The current numbers should therefore be read as what FrameFT achieves for free.

---

> > ### Author Rebuttal · Reviewer_Mkvh · 2026-04-01
> >
> > Most of the doubts have been resolved, and I have decided to keep the score of "weak accept".

---

> > > ### Author Response · Authors · 2026-04-05
> > >
> > > Dear Reviewer Mkvh,
> > >
> > > Thank you for reviewing our rebuttal and your positive assessment of our work. We are glad that we were able to resolve your concerns through our response. We remain available to answer any additional questions during the discussion period.
> > >
> > > Best regards,
> > > The Authors

---

### Official Review · Reviewer_fLfT · 2026-03-12

**Soundness:** 3
**Presentation:** 3
**Significance:** 3
**Originality:** 3
**Overall Recommendation:** 4
**Confidence:** 3

**Summary:**

The paper introduces FrameFT, a novel Parameter-Efficient Fine-Tuning (PEFT) strategy for large foundation models. FrameFT shifts the paradigm of LoRA by modeling the weight update using a sparse coefficient matrix projected into a "Fusion Frame" basis. The paper provides a formal convergence analysis, proving that FrameFT preserves the Lipschitz smoothness of the loss landscape, ensuring stable fine-tuning. Extensive experiments demonstrate that the performance of FrameFT is competitive with or exceeds LoRA and other SOTA methods with fewer parameters than LoRA.

**Compliance With Llm Reviewing Policy:**

Affirmed.

**Final Justification:**

My questions have been adequately addressed, and I have no further comments. Based on the contributions of the paper and the feedback from other reviewers, I would like to raise my recommendation to “Weak Accept.”

**Key Questions For Authors:**

(1) Since the early and late layers of the network typically represent fundamentally different features, does this shared projection bottleneck the expressivity of deep networks?
(2) Given the current lack of hardware support for the specific structured sparsity, how does the actual training time of FrameFT compare to standard LoRA?
(3) How sensitive is the final fine-tuning performance to the specific tight fusion frame generation method used?

**Limitations:**

The authors have adequately addressed the primary technical limitation of their work. The authors have included an Impact Statement that notes that reducing fine-tuning compute requirements reduces energy usage and broadens AI accessibility.

**Strengths And Weaknesses:**

The major strengths include: (1) The paper is well motivated and clearly written. (2) The paper provides a solid mathematical foundation and presents formal proofs on convergence and bounds. (3) The approach achieves SOTA or competition performance with a fraction of the LoRA parameters, making it desirable for real-world deployment.

The weaknesses include: (1) Frame theory is not a standard tool in mainstream deep learning. Readers without a relevant background in functional or harmonic analysis may find it hard to follow. (2) The method relies heavily on structured sparsity to realize the actual benefits of memory and computation. (3) Wide adoption will require community effort to build the specialized kernels needed to fully exploit the sparse coefficient matrix on GPUs.

---

> ### Author Rebuttal · Authors · 2026-03-31
>
> > W1. Frame theory is unfamiliar to general DL audiences.
>
> Frame theory, at least conceptually, is less non-standard than may appear at first: Fourier bases (FourierFT), wavelets, and RoPE positional encodings are all special cases of finite frames. Fusion Frames extend these familiar constructions to structured, overlapping subspaces with formal guarantees that none of the special cases individually tackle. The overhead is bounded: Section 2 gives the three main operators in Frame Theory, and every subsequent result builds directly on these. We included textbook references for readers who want more details. Practitioners who are interested in using FrameFT only need to see that the projection matrices P_m and P_n are generated once by Spectral Tetris and then frozen.
>
> We also note that accessibility concerns are often evaluated against outcomes we get. FrameFT's hyperparameters are fewer and more stable than LoRA. It requires no SVD. In practice, it is simpler to use than the methods it outperforms.
>
> > W2. Structured sparsity is required for memory/compute benefits.
>
> We agree with the observation and would reframe it as a feature rather than a concern. FrameFT's reliance on structured sparsity is analogous to LoRA relying on low-rank structure. This is true and entirely by design. The distinction that matters more for the reader is that FrameFT's sparsity is not an approximation or post-hoc compression. It follows from the properties of the Spectral Tetris construction, so there is no approximation error and no need for threshold tuning.
> The empirical behavior addresses the expressivity issue nicely: across NLU (Table 1), instruction tuning (Table 3), and vision classification (Table 4), FrameFT matches or exceeds methods with far higher parameter counts. A _sparse_ parameterization that outperforms _dense_ alternatives is clearly not constrained by sparsity! The throughput numbers (39.4k vs. 23.6k tokens/sec of LoRA and 1.8k of FourierFT) show us the efficiency dividend with no corresponding expressivity cost.
>
> > Wide adoption requires community wide effort to build special kernels.
>
> We should clarify. Throughput gains in Table 2: FrameFT at 39.4k tokens/sec versus LoRA at 23.6k and FourierFT at 1.8k on Llama-2-7B are achieved entirely with native PyTorch. No custom CUDA kernels were used. Our reported numbers are therefore conservative lower bounds on FrameFT's true efficiency, which comes from the block-sparsity of Spectral Tetris frames (PyTorch already partially exploits it).
>
> On the broader concern, we feel that community effort is no longer a prerequisite. The kernel development landscape has radically changed in the last few months. Tools like Triton and Makora have made writing fused sparse projection kernels a well-scoped engineering task (effort requiring coordinated community support helps but not essential). The block-diagonal structure of C_l and the fixed, predictable sparsity pattern of the Fusion Frame projections are precisely the structures these tools handle most naturally. We will release optimized kernels on the GitHub repo that will offer additional improvements over those reported in the paper.
>
> > Q1. Since early/late features differ, does a shared projection limit expressivity?
>
> Thank you for the nice question! The ablation answers this directly. The table below shows that shared and per-layer random masks perform identically within variance (86.18 ± 0.24 vs. 86.26 ± 0.21 Avg). This confirms that cross-layer sharing does not bottleneck expressivity, aligning with Lemma 3.1: the optimization geometry relies on the rank of the coefficient difference matrix, not on specific nonzero locations.
>
> Surprisingly, adaptive heuristics perform worse. We hypothesize task-relevant subspace interactions are distributed across the coefficient matrix, rather than concentrated in high-magnitude or high-gradient pretrained weights. Random sparsity makes no assumptions and samples this distributed structure more faithfully. We will add this ablation in the revision.
>
> | Method                | sst2             | mrpc             | cola             | qnli             | rte              | stsb             | Avg.             |
> | --------------------- | ---------------- | ---------------- | ---------------- | ---------------- | ---------------- | ---------------- | ---------------- |
> | Shared random mask    | $94.4 \pm 0.17$  | $92.31 \pm 0.46$ | $66.58 \pm 0.77$ | $92.35 \pm 0.12$ | $80.21 \pm 0.82$ | $90.96 \pm 0.11$ | $86.18 \pm 0.18$ |
> | Per-layer random mask | $94.10 \pm 0.16$ | $93.53 \pm 0.32$ | $66.44 \pm 0.61$ | $92.31 \pm 0.14$ | $80.34 \pm 0.39$ | $90.86 \pm 0.09$ | $86.26 \pm 0.17$ |
> | Magnitude-based mask  | $94.19$         | $92.49$          | $65.82$          | $91.61$          | $78.40$          | $91.01$          | $85.60$          |
> | Gradient-based mask   | $89.39$          | $74.59$          | $60.30$          | $84.80$          | $63.49$          | $76.40$          | $74.83$          |

---

> > ### Author Rebuttal · Reviewer_fLfT · 2026-04-03
> >
> > I thank the authors for their detailed response. My questions have been adequately addressed, and I have no further comments. Based on the contributions of the paper and the feedback from other reviewers, I would like to raise my recommendation to “Weak Accept.”

---

> > > ### Author Response · Authors · 2026-04-05
> > >
> > > Dear Reviewer fLfT,
> > >
> > > We thank you for your favorable assessment and for raising your score. We are glad our rebuttal fully resolved your concerns. We remain available to answer any further questions during the discussion period.
> > >
> > > Best regards,
> > >
> > > The Authors.

---

### Decision · Program_Chairs · 2026-04-30

**Decision:**

Accept (regular)

**Comment:**

This paper proposes FrameFT, a parameter-efficient fine-tuning method based on fusion frame theory. Unlike LoRA's low-rank decomposition, FrameFT models parameter updates as a sparse coefficient matrix projected onto a fusion frame basis, generating block-sparse projection matrices via the Spectral Tetris algorithm that can be shared across layers, significantly reducing parameter count and storage footprint.

Technical strengths: On RoBERTa-Base/GLUE, FrameFT (24K parameters) matches LoRA (300K parameters) with the same performance. Storage for Llama-2-7B is only 1.28MB (vs. 67.1MB for LoRA). Inference throughput reaches 39.4k tokens/sec, 1.67× faster than LoRA. Theoretical analysis proves preservation of Lipschitz smoothness, ensuring stable convergence. Experiments cover NLU, instruction tuning, and vision classification tasks, comparing with multiple SOTA methods.

Limitations and responses: Performance gains primarily manifest in parameter efficiency rather than absolute accuracy improvements; frame theory may present a high barrier for some readers; experimental reporting had some incompleteness (e.g., missing full fine-tuning baselines, unreported throughput variance). The authors addressed these concerns in the rebuttal by providing missing experiments (full fine-tuning results, multi-seed variance, scaling factor sensitivity analysis, extended coefficient ablation). All reviewer concerns have been resolved.

All four reviewers gave weak accept. This method provides a novel theoretical perspective and efficient implementation for the PEFT field.